# Modifying Factors of Adult Hippocampal Neurogenesis: A Dorsoventral Perspective in Health and Disease

**DOI:** 10.3390/cells15010059

**Published:** 2025-12-29

**Authors:** Ioannis Erginousakis, Costas Papatheodoropoulos

**Affiliations:** Laboratory of Physiology-Neurophysiology, Department of Medicine, University of Patras, 265 04 Patras, Greece; up1084168@ac.upatras.gr

**Keywords:** adult hippocampal neurogenesis, ventral hippocampus, dorsal hippocampus, dorsoventral, neuroplasticity, neurodegenerative disease, psychiatric disease, neuromodulation

## Abstract

Adult hippocampal neurogenesis (AHN) is a dynamic process that sustains neural plasticity and contributes to cognition, emotion, and stress resilience. While its functional significance in humans remains debated, growing evidence suggests that AHN plays an important role in health and disease. In this review, we summarize intrinsic and extrinsic factors that modulate AHN, with particular emphasis on hormones, behavior, diet, and their impact along the hippocampal dorsoventral axis, where baseline neurogenesis is higher dorsally, but ventral neurogenesis exhibits greater plasticity and sensitivity to modulatory systems. We highlight how cognitive stimulation, physical activity, and rewarding experiences preferentially enhance dorsal hippocampal neurogenesis, whereas chronic stress and glucocorticoids mainly impair neurogenesis in the ventral hippocampus. Nutritional influences such as caloric restriction, high-fat diets, vitamins, and polyphenols are also considered, with evidence for region-specific effects. We further examine the relevance of AHN alterations in neuropsychiatric diseases, such as major depressive disorder, schizophrenia, Alzheimer’s disease, and addiction, highlighting both common mechanisms and disorder-specific vulnerabilities. Collectively, current findings suggest that AHN serves as a converging pathway connecting lifestyle, neuroendocrine regulation, and psychiatric or neurodegenerative disease. Recognizing the dorsoventral specialization of AHN could refine mechanistic models of brain function and inform the development of targeted and distinct therapeutic strategies for cognitive and affective diseases.

## 1. Introduction

Neurogenesis, the process by which new neurons are formed within the brain, occurs both during embryonic development and, to a more limited extent, in the adult brain. Early autoradiographic studies by Altman & Das provided the first evidence of adult neurogenesis in the mammalian brain [1,2].

Although recent literature remains divided, with some studies supporting [3,4,5,6,7,8,9] and others questioning [10,11,12] the extent and significance of human adult hippocampal neurogenesis (AHN) (for review, see [13]), its potential relevance for hippocampal function and neuropsychiatric disease remains an active area of investigation. In rodents, which provide the most detailed quantitative and mechanistic data on adult neurogenesis, the hippocampus consistently exhibits the highest rates [14]. In rodent models, growing evidence indicates that this neurogenic capability is not uniformly distributed within the hippocampus but instead follows a longitudinal (dorsoventral or septotemporal) gradient. The dorsal hippocampus is primarily associated with spatial, cognitive, and mnemonic functions, whereas the ventral hippocampus is more strongly involved in emotional regulation, stress responses, and affective behavior. AHN mirrors this specialization: baseline proliferation tends to be higher dorsally, while the ventral dentate gyrus is more sensitive and plastic in response to hormonal, metabolic, and environmental modulation. Introducing this organizational axis early provides an essential framework for understanding how intrinsic and extrinsic factors exert region-specific influences on both cognition and emotion.

The hippocampus subserves a wide range of functions, including spatial navigation and cognition [15,16], learning and memory consolidation [15,17] and emotional processing [18]. AHN occurs in the subgranular zone (SGZ) of the hippocampal dentate gyrus (DG), where neural stem/progenitor cells give rise to newborn granule neurons that migrate into the granule cell layer and subsequently integrate into existing hippocampal circuits [19], linking experience to structural plasticity. Region-specific regulatory influences on AHN have been increasingly recognized [20], and these dorsoventral distinctions form a central framework for the present review.

The discovery of AHN and its capacity to modulate neural plasticity has prompted intensive investigation into how intrinsic and extrinsic factors modulate its levels. Modifiable influences, such as hormonal environment, behavior, and diet, are of particular interest due to their potential to reveal molecular mechanisms through which lifestyle impacts brain function. Notably, several studies indicate that some of these factors appear to differentially affect AHN along the hippocampal axis, suggesting that both baseline neurogenesis and regional plasticity may determine the direction and magnitude of their effects. These cellular processes provide the framework for understanding how intrinsic and extrinsic factors differentially modulate adult hippocampal neurogenesis, including region-specific effects along the dorsoventral axis.

While numerous reviews have summarized modulators of AHN [21,22,23], the subregional perspective has remained relatively overlooked. The goal of this review is therefore to synthesize evidence on the most widely studied modifiers of AHN in animal models and humans, with explicit attention to subregion-specific effects along the hippocampal dorsoventral axis. We further examine how these mechanisms relate to neuropsychiatric disease, highlighting their potential translational and therapeutic implications. To maintain a focused narrative, the present review emphasizes findings most relevant to the dorsoventral specialization of the hippocampus and highlights species-specific considerations where they influence interpretation.

In selecting modulatory factors, we focused on those most widely studied across animal and human research. For several of these factors, sufficient data exist to evaluate dorsoventral differences, while for others, subregional analyses are not yet available. We therefore present dorsoventral distinctions whenever supported by evidence and summarize global effects where such comparisons have not been performed.

Furthermore, altered AHN has been associated with psychiatric conditions such as major depression and schizophrenia, as well as with neurodegenerative diseases including Alzheimer’s disease. These observations motivate a unified examination of how dorsoventral neurogenesis may differentially contribute to clinically distinct yet hippocampus-related disorders.

## 2. Adult Hippocampal Neurogenesis in Humans

Although this section summarizes general features of AHN, these mechanisms form the basis for understanding how modulatory influences operate differently along the dorsoventral axis. These cellular processes provide the framework for understanding how intrinsic and extrinsic factors modulate adult hippocampal neurogenesis at distinct developmental stages.

Despite the increasing interest and research being conducted on AHN over the past decades, there is still notable controversy over whether its presence is significant enough in humans, with some claiming that it is purely an evolutionary remnant, especially in organisms with high cognitive function [12]. However, there have been some studies which support that a considerable level of AHN does in fact exist in humans. Eriksson et al. provided early evidence of AHN by identifying BrdU-labeled neurons in the human hippocampus [24].

Spalding et al. later used atmospheric ^14^C incorporation as a birth-dating method and estimated substantial daily neuronal turnover in the adult DG [25], although this interpretation remains debated [12]. Nevertheless, even if AHN levels in humans are low, clarifying the underlying mechanisms may still provide insights with translational relevance.

Adult hippocampal neurogenesis occurs when quiescent neural stem and progenitor cells in the SGZ become activated and proceed through a sequence of proliferation, neuronal differentiation, migration into the granule cell layer, maturation, synaptic integration, and long-term survival, based on the framework established by Kempermann and colleagues [26,27,28] (Figure 1). These stages have been defined primarily in rodents, where direct experimental evidence is available. The molecular markers described in this section are used experimentally to identify cells at different stages of this trajectory and do not themselves represent mechanisms.

Across these stages, NSCs give rise to transiently amplifying progenitors and subsequently to neuroblasts, which migrate into the granule cell layer and begin synaptic integration. A substantial proportion of newborn neurons undergo programmed cell death during early survival, and their retention is highly sensitive to physiological state.

Immature granule neurons display heightened excitability and reduced inhibition, which lowers the threshold required to induce long-term potentiation (LTP). A lower LTP threshold allows synaptic strengthening to occur more readily, thereby contributing to the greater synaptic plasticity characteristic of neurons at this stage. Those that survive progress into the late maturation phase, during which they establish stable connections within existing hippocampal circuits [26].

While AHN in rodents is robust and experimentally tractable, evidence in humans remains more limited and methodologically constrained. Thus, conclusions drawn from rodent studies provide mechanistic insight but do not necessarily reflect the magnitude or dynamics of AHN in the adult human hippocampus.

Because modulatory influences can act at different phases of AHN, altering proliferation, early survival, maturation, or long-term integration, we refer to immature, mature, and apoptotic neurons to distinguish these functionally relevant stages. This allows us to describe more precisely how hormonal, behavioral, and dietary factors affect the neurogenic trajectory.

## 3. Factors Modifying Hippocampal Neurogenesis

In the following sections, we summarize the major intrinsic and extrinsic modulatory factors of AHN, highlighting their relevance to dorsoventral specialization when data are available. It is important to note that most modulatory influences described in this section are obtained from rodent studies, whereas corresponding evidence in humans is scarcer and often indirect. Species differences should therefore be considered when interpreting the degree and direction of AHN modulation.

### 3.1. Hormones

Hormones are soluble chemical messengers produced and secreted by endocrine glands in multicellular organisms. They contribute to homeostasis by acting in multiple organs, such as the brain. Their action is mediated through intracellular or extracellular receptors which are expressed at certain target cells. Research over the past years has underscored the neurogenic impact of various hormones in the hippocampus, which could involve stimulatory, inhibitory or neuroprotective outcomes. Although circulating hormone levels are not directly regulated by the hippocampal neurogenic area, the local sensitivity to these hormones (through changes in receptor expression, intracellular signaling pathways, or modulatory inputs) can be altered by experience, stress, and metabolic state.

#### 3.1.1. Cortisol/Corticosterone

Most findings summarized below come from rodent studies, where the effects of corticosterone on AHN have been examined experimentally. Cortisol in human and corticosterone in rodents, collectively referred to as CORT, are steroid hormones synthesized by the zona fasciculata of the adrenal glands and released through activation of the hypothalamic–pituitary–adrenal axis, typically in response to stress. As steroid hormones, they primarily exert their actions through intracellular glucocorticoid receptors (GRs) or mineralocorticoid receptors (MRs), which regulate gene transcription through DNA-binding or interaction with transcription factors such as NF-κB, STAT3, and AP-1.

MR and GR are abundantly expressed in hippocampal neurons, with MRs exhibiting ~10-fold higher affinity for CORT than GRs [29,30]. Pharmacological studies indicate that MR agonists and GR antagonists promote hippocampal neurogenesis [31,32,33], whereas chronic CORT exposure reliably suppresses AHN in rodents [34,35,36,37,38]. Notably, the ventral hippocampus appears more intrinsically sensitive to glucocorticoid-mediated inhibition than the dorsal hippocampus [39]. In rodent studies, acute CORT exposure refers to brief elevations lasting minutes to hours, while chronic exposure denotes sustained increases across days or weeks. These temporal patterns engage distinct receptor-mediated pathways, with chronic elevations preferentially engaging GR-induced anti-neurogenic mechanisms. Recent work further suggests that CORT-induced deficits in AHN may involve p21 overexpression and associated ROS accumulation [34].

High and sustained CORT levels suppress AHN, whereas transient CORT elevations during activities such as exercise, enrichment, sexual behavior, and learning can enhance AHN through concurrent activation of reward and neurotrophic pathways. These experiences activate reward circuitry, neuromodulatory systems, and neurotrophic pathways (e.g., BDNF) that counterbalance the suppressive effects of glucocorticoids [22,40,41,42,43,44,45,46]. Thus, CORT elevation alone does not determine AHN outcomes; instead, the physiological context in which CORT rises is critical. Together, the available findings suggest a U-shaped relationship between circulating CORT levels and AHN, in which very low or sustained high glucocorticoid levels are detrimental, whereas moderate and transient elevations can support neurogenesis by engaging MR-dominant signaling [31,32].

Taken together, chronically elevated CORT, whether due to persistent stress or exogenous administration, tends to suppress AHN, with higher vulnerability seen in the ventral hippocampus [39]. In contrast, low-to-moderate CORT elevations may support AHN, likely through MR-dominant signaling, whereas sustained high CORT engages GR-mediated anti-proliferative pathways [47].

Activities that produce transient CORT elevations may enhance AHN by maintaining a favorable MR/GR balance [22,40,41,42,43,44,45,46], whereas conditions that produce chronic CORT elevations [48] can impair AHN through GR saturation and downstream inhibition of progenitor proliferation [29,30,31,32,33,47]. Further work is needed to delineate how varying CORT levels modulate neurogenesis in the dorsal versus ventral hippocampus. Overall, glucocorticoid-induced suppression of AHN is not uniform along the dorsoventral axis: the ventral DG, involved in emotional and stress regulation, shows greater vulnerability to corticosterone elevation, whereas the dorsal DG shows comparatively milder and more transient changes. Figure 2 depicts corticosterone as a representative hormonal modulator because its effects on AHN and its stronger impact on the ventral DG are among the best characterized in rodent studies.

#### 3.1.2. Estrogens

Estrogens are a group of steroid hormones that are known for their major role in the development of the female reproductive system. Outside of this traditional function, estrogens also have considerable effects on multiple organs and tissues, such as the bone, liver, blood vessels, skin, fat and central nervous system. The three most dominant types of estrogen are 17β-estradiol, estrone and estriol. 17β-estradiol is mainly produced by the ovaries, estriol is produced by the placenta and estrone is produced in adipose tissue via aromatization. Estrogens primarily exert their action through intracellular nuclear receptors (ERα and ERβ), but also through the membrane-bound receptors G protein-coupled estrogen receptor 1 (GPER1), ER-X and Gq-mER [49].

ERα, ERβ and GPER1 receptors are expressed widely in the brain, with ERβ mainly being found in the hippocampal DG [50]. GPER1 deficiency has been shown to impair neurogenesis in the hippocampus, indicating that GPER1 could play a role in the induction of hippocampal neurogenesis [51]. Multiple researchers have noted that estradiol stimulates AHN and cell proliferation of adult female rats in a time- and dose-dependent manner [52,53]. The dose of estradiol that has been most associated with positive effects in hippocampal neurogenesis is 10 μg. Acute administration of estradiol has been correlated with increased cell proliferation and survival in the hippocampus, while chronic administration has been associated with the opposite effects [53,54,55]. Research regarding estrogen’s impact on the male hippocampus, however, has been scarce. There are indications that survival of young granule neurons in the male DG is increased after estradiol administration [56,57]. Many authors also highlight the fact that even though most experiments are being conducted with estradiol, which is a potent estrogen, different types of estrogen can also have different effects on neurogenesis. Estrone administration, for example, has been associated by many with a decreased neurogenic potential, especially when it comes to cell proliferation [53,55]. In addition, aromatase inhibition has been reported to augment AHN levels in middle-aged female mice [58].

Regarding the regional impact of estrogens on AHN, available evidence remains limited. In one study, the phytoestrogen daidzein enhanced neurogenesis in both the ventral and dorsal hippocampus, with more prominent effects dorsally [59]. Overall, estrogens tend to transiently enhance AHN in female rodents, whereas prolonged or chronic exposure may reduce hippocampal cell proliferation, suggesting a complex, time-dependent regulation. The negative impact of chronic estrone administration also needs further evaluation, as this hormone is elevated in obesity due to increased aromatization of androgens in adipose tissue [60]. Taken together, these findings indicate that estrogens exert regionally distinct influences on AHN along the hippocampal longitudinal axis. Experimental data suggest that estrogen-induced proliferation and neuronal differentiation are generally stronger in the dorsal DG, consistent with its cognitive specialization, whereas ventral neurogenesis appears less robust or more transiently modulated, potentially reflecting differences in ER distribution and stress-related signaling between the two poles of the hippocampus.

#### 3.1.3. Androgens

Androgens are a group of steroid hormones responsible for the development of the male reproductive system. They also have anabolic actions in the musculoskeletal system. The main androgens found in mammals are testosterone and Dihydrotestosterone. Testosterone is produced and secreted by Leydig cells in the testis and can be converted to DHT by 5α-reductase in peripheral tissue. Androgens mainly exert their actions by binding to intracellular Androgen Receptors (ARs) and modulating the expression of multiple genes in the target cells.

ARs are expressed in multiple regions of the brain, such as the hippocampus [61]. Many researchers report a positive correlation trend between long-term androgen administration and the survival of new hippocampal neurons [62,63] in both the dorsal and the ventral hippocampus, with statistically significant benefit being observed only in young male rats [61,62,64]. On the contrary, anabolic steroid androgens, which are commonly used among athletes, have been shown to abolish the positive neurogenic effects of exercise [65]. Overall, androgens, with the exception of anabolic steroids, seem to be beneficial to AHN in an age and sex specific pattern, favoring only young males and only if administered chronically.

Evidence on the dorsoventral specificity of androgen effects on AHN remains limited, although some findings suggest that androgenic modulation may exhibit regional differences. Testosterone and its metabolites appear to enhance cell proliferation and neuronal survival predominantly in the dorsal DG [62,63], consistent with its role in cognitive and spatial processes, whereas effects in the ventral region are weaker or context-dependent [66]. This gradient may reflect variations in AR density, local aromatization to estrogens, and the distinct functional circuitry of the dorsal and ventral hippocampus. For instance, AR expression is somewhat higher in the dorsal hippocampus [66,67], suggesting greater responsiveness in this region. Conversely, alterations in androgen/estrogen balance, such as those induced by aromatase inhibition, can enhance ventral neurogenesis [58]. Together, these findings indicate that androgenic modulation acts in a region-dependent and context-sensitive manner along the hippocampal longitudinal axis.

#### 3.1.4. Oxytocin

Oxytocin is a peptide hormone mainly synthesized in the hypothalamus and secreted into the bloodstream by the posterior pituitary gland. Acting through its transmembrane G-protein-coupled receptor (OXTR), oxytocin regulates multiple physiological and behavioral functions, including parturition, uterine contraction, milk ejection, sexual behavior, and social interaction [68,69].

OXTR is abundantly expressed in the hippocampus of rodents but appears much less prevalent in the human hippocampus [70,71].

Research on the impact of oxytocin on AHN remains limited, though several studies indicate that chronic intranasal or peripheral oxytocin administration increases neurogenic activity, particularly in the ventral DG [72,73,74]. Peripheral oxytocin treatment can even counteract the inhibitory effects of stress and glucocorticoids on hippocampal cell proliferation [64]. However, other studies reported reduced neurogenesis following oxytocin administration under acute treatment paradigms [75], suggesting that the duration and route of administration critically determine its neurogenic outcome.

Evidence on the dorsoventral specificity of oxytocin’s effects on AHN remains limited. In adult rats, systemic or intracerebral oxytocin administration promotes social interaction and stress resilience but exerts modest or inconsistent effects on hippocampal cell proliferation overall [75]. Nevertheless, the behavioral domains influenced by oxytocin, such as social bonding, anxiety regulation, and affective processing, are functions predominantly mediated by the ventral hippocampus [76,77]. Neurogenesis in the ventral DG has been linked to the modulation of stress-related and emotional behaviors [78], suggesting that oxytocin’s neuromodulatory actions may preferentially influence this hippocampal segment. Although direct comparisons of dorsal and ventral neurogenic responses to oxytocin are scarce, current evidence supports a model in which ventral AHN is particularly sensitive to oxytocin-mediated regulation, consistent with the peptide’s broader role in emotional adaptation and social cognition.

#### 3.1.5. Thyroid Hormones

Thyroid hormones (THs), primarily triiodothyronine (T3) and thyroxine (T4), are peptide hormones synthesized and secreted by the thyroid gland. They exert their effects through intracellular and membrane-associated thyroid hormone receptors (TRs), including TRα1, TRα2, and TRβ1, which are widely expressed in the adult brain [79,80]. THs play essential roles in regulating metabolism, homeostasis, and neurodevelopment.

Experimental models indicate that hypothyroid states reduce the survival and differentiation of adult DG cell progenitors in both mice and rats [81,82,83]. In contrast, hyperthyroidism has been associated with enhanced neuronal differentiation and postmitotic progenitor survival in mice [82], though such effects were not replicated in rats [81], suggesting species-dependent variability. To date, no studies have explicitly examined the impact of THs on neurogenesis along the dorsoventral hippocampal axis. Nevertheless, indirect evidence suggests potential region-specific effects. TRα and TRβ are differentially distributed across hippocampal subregions [84].

Moreover, TH signaling interacts with neurotrophic and glucocorticoid pathways [85,86], both of which show stronger regulatory impact in the ventral hippocampus. This suggests that thyroid hormones may influence dorsal neurogenesis more directly through receptor-mediated transcription, while affecting ventral neurogenesis indirectly through stress-related endocrine interactions.

In summary, hypothyroidism appears to impair, and hyperthyroidism to potentially enhance, AHN, but species differences and methodological variability complicate interpretation. Future research should clarify how TH receptor distribution and signaling intersect with dorsoventral specialization to determine regional susceptibility or plasticity within the adult hippocampus, as well as their relevance to human neurogenesis.

#### 3.1.6. Melatonin

Melatonin is an indoleamine hormone primarily synthesized and secreted by the pineal gland, typically in response to darkness. It regulates circadian rhythm, core body temperature, energy metabolism, and immune function, and also exerts potent antioxidant and anti-inflammatory effects. Melatonin acts mainly through its G-protein-coupled receptors MT1 and MT2, which are distributed throughout the mammalian brain, including the hippocampus. MT1 receptors are abundantly expressed across the hippocampus, whereas MT2 receptors show higher density in the CA3 region [87]. It is important to note that commonly used laboratory strains such as C57BL/6 mice produce little to no endogenous melatonin due to a mutation in the arylalkylamine N-acetyltransferase gene, a factor that should be considered when interpreting melatonin-related findings in rodent models.

Several studies have investigated melatonin’s role in AHN. Pinealectomy in rats markedly decreases the number of newborn hippocampal neurons, an effect reversed—and even surpassed—by chronic melatonin supplementation [88,89]. Consistent findings demonstrate that melatonin primarily enhances neuronal survival and differentiation, rather than cell proliferation, thereby potentiating the effects of other pro-neurogenic stimuli such as physical exercise [89,90,91,92]. Melatonin’s neuroprotective role is further supported by studies showing attenuation of AHN deficits induced by hyperglycemia, glucocorticoid exposure, or manganese toxicity (excessive Mn exposure causing oxidative stress and neurotoxic effects) [93,94,95]. Additionally, melatonin administration improves recovery after ischemic injury, promoting neuronal regeneration possibly through MT2-mediated signaling [96,97].

Collectively, these findings establish melatonin as a robust enhancer of hippocampal neurogenesis and neuronal resilience. Its beneficial effects appear to stem largely from antioxidant and anti-inflammatory actions that improve survival and maturation of newborn neurons.

Regarding dorsoventral specificity, evidence points to a ventral predominance of melatonin’s neurogenic and behavioral effects. In rodents, melatonin treatment ameliorates stress-induced anxiety and depression-like behaviors, functions mediated primarily by ventral hippocampal circuits [77,98]. Moreover, activation of MT2 receptors has been shown to promote AHN and antidepressant-like outcomes through upregulation of Brain-Derived Neurotrophic Factor (BDNF) and modulation of glucocorticoid signaling [99,100]. Although direct dorsal–ventral comparisons remain limited, these data suggest that melatonin preferentially enhances ventral DG plasticity, consistent with its role in emotional regulation and stress resilience. Future work should aim to clarify whether melatonin’s pro-neurogenic actions extend equivalently to dorsal hippocampal domains involved in cognitive processes.

#### 3.1.7. Growth Hormone, IGF-1

Growth Hormone (GH) is a peptide hormone synthesized, stored and secreted by somatotropic cells in the anterior pituitary gland. It plays a major role in somatic growth, regulating bone and organ development, as well as lipid and mineral metabolism. Although GH acts directly on tissues through growth hormone receptors (GHRs), many of its effects are mediated indirectly by insulin-like growth factor-1 (IGF-1). GH stimulates the synthesis of IGF-1 in the liver and peripheral tissues, and IGF-1 in turn promotes tissue growth via its membrane-bound receptor tyrosine kinase (IGF1R).

Both GHRs and IGF1Rs are expressed in multiple brain regions, including the hippocampus [101,102]. Peripheral administration of GH or IGF-1 increases cell proliferation in the hippocampal formation [103,104,105,106], and several studies suggest that exercise-induced neurogenesis is partly mediated by upregulation of circulating GH and IGF-1 [107,108]. These trophic hormones enhance neurogenesis by promoting angiogenesis, synaptic plasticity, and neuronal survival through BDNF- and Vascular Endothelial Growth Factor (VEGF)-dependent pathways [109,110,111].

To date, no studies have directly examined the impact of GH or IGF-1 on neurogenesis across the dorsoventral axis of the hippocampus. However, indirect evidence suggests region-specific modulation. Exercise and growth factor signaling have been shown to preferentially engage ventral hippocampal circuits associated with stress regulation and emotional processing [77,98], implying that GH/IGF-1–mediated neurogenesis may be more dynamically regulated in this region. Conversely, the dorsal hippocampus, enriched in GH receptor expression, could support GH-driven enhancement of spatial and cognitive processes [112,113].

Collectively, GH and IGF-1 appear to positively modulate hippocampal cell proliferation and survival, but their region-specific influence along the hippocampal longitudinal axis remains to be determined. Future studies should address whether these hormones differentially regulate ventral (emotional) and dorsal (cognitive) neurogenesis, particularly in the context of exercise and aging.

#### 3.1.8. Erythropoietin

Erythropoietin (EPO) is a peptide hormone that is mainly synthesized in the kidneys in response to hypoxia and stimulates erythropoiesis (proliferation and differentiation of erythrocyte precursors). It exerts its actions via its membrane receptors (EPO-R) which are expressed at target cells.

EPO-R expression is observed in various brain cell types, including hippocampal neural progenitor cells [114]. There are reports indicating that EPO enhances neurogenesis and inhibits apoptosis of newborn cells in the hippocampal DG [115,116,117]. Other researchers have observed increased rates of neurogenesis correlated with increased levels of EPO due to cognitive-challenge-induced hypoxia in hippocampal neurons [118]. Furthermore, the impact of EPO in neurogenesis could suggest another mechanism through which the positive neurogenic effects of exercise are mediated [119]. To our knowledge, no studies have yet explored EPO’s effect on neurogenesis in the ventral compared to the dorsal hippocampus.

Overall, there has been some evidence that supports a positive neurogenic effect of EPO in the hippocampus, but further research could explore its exact action along the hippocampal longitudinal axis.

### 3.2. Lifestyle and Environmental Factors

#### 3.2.1. Physical Activity

Most mechanistic evidence for activity-induced AHN enhancement derives from rodent studies, whereas human data remain more limited and indirect. Physical activity has long been associated with improved brain health and cognitive performance. Seminal work by van Praag et al. (1999) [120] demonstrated that voluntary running robustly increases progenitor proliferation and survival of newborn neurons in the adult rodent dentate gyrus. In humans, even light aerobic exercise has been shown to improve multitasking and executive function [121], and chronic intermittent voluntary running in mice produces long-lasting benefits on learning and memory [122]. These converging findings have prompted extensive investigation into the molecular- and circuit-level mechanisms through which physical activity promotes neurogenesis and cognition.

Human imaging studies reveal that regular aerobic activity in late adulthood increases anterior hippocampal gray matter volume, a region enriched in AHN, suggesting that the beneficial cognitive effects of exercise may be mediated through augmented neuronal production [123,124]. In both rodents and humans, physical activity elevates circulating growth factors such as IGF-1, BDNF, and VEGF [125,126], which together promote synaptic plasticity, angiogenesis, and neurogenesis in the hippocampal formation [41,108,120]. This effect may depend on serotonin, since serotonin-deficient mice failed to exhibit hippocampal neurogenesis enhancements following physical activity [127]. More recently, irisin, a myokine secreted by skeletal muscle during exercise, has been shown to exert anti-inflammatory and neuroprotective effects, contributing to activity-induced hippocampal neurogenesis [128,129]. The adipokine adiponectin may similarly support the pro-neurogenic and cognitive benefits of physical exercise [130]. Furthermore, ablation of newborn neurons following running abolishes the typical exercise-induced improvements in cognition and memory [131], demonstrating that AHN plays a causal role in mediating exercise-driven brain plasticity. It should be noted, however, that excessive or prolonged voluntary running can sometimes reduce neurogenesis, possibly due to increased physiological stress and glucocorticoid release [132].

Regarding the distribution of exercise-induced neurogenesis along the hippocampal dorsoventral axis, most studies report stronger effects in the dorsal DG, consistent with its role in spatial learning and cognitive processing [41,90,133]. For example, environmental enrichment combined with voluntary exercise preferentially enhanced dorsal neurogenesis, and this increase correlated with improved spatial performance, whereas ventral neurogenesis contributed more prominently to stress resilience and affective behavior [134]. Other studies, however, observed increases in both dorsal and ventral hippocampal segments, with voluntary running preferentially enhancing ventral DG neurogenesis through IGF-1 receptor signaling [127], suggesting that exercise engages distinct proliferating progenitor pools depending on intensity, duration, and hormonal state. Overall, physical activity enhances AHN through systemic and local trophic factors, with dorsal neurogenesis predominantly supporting cognitive gains and ventral neurogenesis mediating emotional regulation and stress adaptation. Clarifying the parameters that shift this balance may help optimize exercise-based interventions for cognitive and affective disorders.

#### 3.2.2. Sleep

Findings summarized below are primarily from rodent models, with limited but supportive evidence from human studies. Sleep is divided into Rapid Eye Movement (REM) and non-Rapid Eye Movement Non-REM (NREM) sleep. REM sleep, comprising 25% of total sleep time, is characterized by random rapid movement of the eyes, with beta waves being recognized on electroencephalogram (EEG) recordings. NREM is further divided into Stage 1 (N1), Stage 2 (N2) and Stage 3 (N3). N1 is the lightest sleep stage and comprises 5% of total sleep time, with theta waves being observed on EEG. N2 sleep is deeper than N1, comprising 45% of total sleep time, and sleep spindles along with K complexes can be seen on EEG. Finally, N3 is the deepest non-REM sleep, comprising 25% of total sleep time, and the EEG at this stage consists of delta waves which have the lowest frequency and the highest amplitude [135].

Adequate sleep is essential for optimal brain function, influencing cognition, memory, and mood regulation [136,137,138]. Since these processes are closely linked to AHN, numerous studies have examined how sleep and neurogenesis interact. Sleep deprivation, fragmentation, or restriction consistently reduce neurogenic potential, impairing newborn neuron proliferation, survival, and differentiation [139] (for review, see [23]). Prolonged or repeated sleep loss has been shown to decrease neurogenesis and cell proliferation in both the dorsal and ventral DG [140]. Furthermore, the decrease in DCX-labeled cells was more pronounced in the ventral hippocampus than in the dorsal region [140].

However, as sleep deprivation also constitutes a potent stressor, elevated glucocorticoid levels may mediate part of the observed neurogenic suppression [141]. Distinguishing direct effects of disrupted sleep from stress-related mechanisms therefore remains an important experimental challenge.

Beyond general suppression, sleep and its disruption appear to influence AHN differentially along the dorsoventral hippocampal axis. While studies in rodents have shown that sleep deprivation preferentially impairs neurogenesis in the ventral DG, corresponding with increases in anxiety-like behaviors [142], it has also been reported that REM sleep deprivation can also reduce cell proliferation in the dorsal DG and impair hippocampal-dependent spatial memory [143].

These findings suggest that sleep disturbances may predominantly affect ventral neurogenesis [140,141], consistent with the ventral hippocampus’s sensitivity to stress and emotional regulation. This is further supported by findings presented by O’Leary and Cryan [77], as well as by Tanti and Belzung [144], who provide comprehensive reviews demonstrating that stress affects multiple stages of AHN preferentially in the ventral rather than the dorsal hippocampus. Additionaly, Perera et al. [145] showed in non-human primates that chronic stress reduced immature neurons in the anterior (ventral equivalent) but not posterior hippocampus, and this effects was correlated with stress-induced anhedonia.

Future studies should employ experimental designs that disentangle the direct neurogenic effects of sleep architecture from those mediated by stress hormones. Such differentiation is crucial, as therapeutic strategies targeting glucocorticoid signaling may protect ventral neurogenesis, while interventions enhancing sleep quality and REM integrity could preferentially restore dorsal and ventral neurogenic balance.

#### 3.2.3. Learning

Experimental data on learning-related modulation of AHN come mainly from rodent studies, where learning paradigms can be precisely controlled. The process of learning has long been associated with the hippocampal formation, and since high neurogenic activity occurs in this region, AHN has been proposed as a mechanism supporting learning and memory. Learning tasks enhance survival of newborn neurons [146].

The relationship between learning and neurogenesis appears to be bidirectional. Inhibition of neurogenic potential in animal models disrupts contextual learning and memory performance [147,148], while specific learning tasks enhance AHN by increasing the survival of newborn granule cells [149]. Training paradigms such as trace conditioning, stimulus contiguity conditioning, long-delay conditioning, and the Morris water maze are associated with elevated neurogenic rates [150]. Moreover, the complexity of the task and the degree of hippocampal engagement correlate positively with the survival and integration of new neurons [150,151,152].

The increased survival of newborn hippocampal neurons has been linked to improved memory precision and cognitive flexibility. For example, in the Morris water maze, mice with enhanced neurogenesis exhibit more spatially precise navigation and greater adaptability when the hidden platform is relocated, reflecting improved memory updating and reduced perseveration [153].

Although most data derive from rodent studies, evidence suggests similar adaptive processes in humans. Neuroimaging and neuropsychological assessments of London taxi drivers, who undergo extensive spatial training, revealed increased posterior hippocampal gray matter volume compared with controls and bus drivers [154,155]. This structural enlargement corresponded to superior performance in spatial navigation and landmark proximity judgments but reduced recall of complex non-spatial visual patterns, indicating a potential functional specialization and trade-off in hippocampal processing. These findings support the concept that persistent cognitive demands can induce hippocampal remodeling—possibly involving AHN—favoring memory improvements in domain-relevant circuits.

Regarding the dorsoventral axis, converging evidence indicates that learning-related increases in neurogenesis are more pronounced in the dorsal hippocampus, which is primarily involved in spatial and contextual learning [134,156] (Figure 3). In contrast, ventral hippocampal neurogenesis appears to contribute more to emotionally charged or reward-based learning [157]. For instance, learning paradigms with strong emotional valence or stress components, such as fear conditioning or reward anticipation, recruit ventral DG circuits where neurogenesis facilitates adaptive behavioral responses [158]. Thus, dorsal neurogenesis supports cognitive precision and spatial mapping, whereas ventral neurogenesis underlies motivational and affective learning, reflecting complementary but distinct roles along the hippocampal longitudinal axis.

#### 3.2.4. Rewarding Experience

Reward-related experiences constitute a major behavioral and motivational state that modulates AHN through dopaminergic and limbic pathways, often in opposition to the effects of stress. Most mechanistic insights into reward-induced modulation of AHN come from rodent studies, where reward paradigms can be experimentally controlled. The reward system comprises a network of interconnected brain regions responsible for evaluating, predicting, and anticipating rewarding stimuli, as well as motivating behavior and generating the sensation of pleasure. The orbitofrontal cortex (OFC) and anterior cingulate cortex (ACC) compute the expected value of potential rewards and the effort required to obtain them. This information is integrated by the ventromedial (vmPFC) and dorsolateral prefrontal cortex (dlPFC) to guide decision-making on reward pursuit [159]. When rewards are obtained, dopaminergic neurons in the ventral tegmental area (VTA) signal deviations between expected and actual outcomes (reward prediction errors) [160,161]. If outcomes are as good as or better than predicted, these neurons increase firing toward limbic and cortical targets, including the prefrontal cortex, cingulate gyrus, nucleus accumbens (NAc), hippocampus, and amygdala. Dopaminergic input to the amygdala and cingulate gyrus generates the emotional experience of pleasure, while projections to the hippocampus facilitate long-term memory consolidation of rewarding events, enabling their future recall [161].

Although research directly investigating the effects of rewarding experiences on AHN is still limited, available data suggest that repeated engagement in rewarding behaviors can enhance neurogenic activity. For example, sexual experience, which robustly activates the reward system [162], increases neuronal proliferation in the hippocampal DG of both young and middle-aged rats [163]. These effects partially restore age-related declines in neurogenesis and improve performance in cognitive tasks [164].

Beyond behavioral experiences, dopamine, the principal neurotransmitter in reward processing, exerts direct neurogenic effects. Dopaminergic stimulation enhances the proliferation of neural precursor cells and the survival of newborn neurons in the DG [165,166], likely via D1/D2 receptor-mediated modulation of BDNF and CREB signaling cascades. This suggests that reward-related dopaminergic activity may couple motivational states to hippocampal structural plasticity.

Although no studies have systematically examined dorsoventral differences in reward-related AHN, converging evidence supports a ventral predominance. The ventral hippocampus is functionally connected to the NAc and amygdala, and neurogenesis in the ventral DG has been shown to modulate reward sensitivity, stress resilience, and affective regulation [157,158]. In contrast, dorsal hippocampal neurogenesis contributes more to cognitive flexibility and spatial memory, which can also influence reward learning indirectly [134]. These findings collectively suggest that rewarding experiences, as well as their dopaminergic reinforcement, may primarily recruit ventral hippocampal neurogenic circuits, aligning with the emotional and motivational functions of this subregion. Future research should explore how distinct rewarding stimuli (e.g., social interaction, novelty seeking, or food-related rewards) differentially influence dorsal and ventral neurogenesis and whether these effects contribute to adaptive versus maladaptive motivational behaviors.

#### 3.2.5. Stress

Stress is one of the most potent negative regulators of AHN, acting primarily through glucocorticoid signaling and HPA-axis activation, representing a fundamental behavioral and physiological state relevant to AHN modulation. The majority of mechanistic evidence derives from rodent models, although human imaging and endocrine data support convergent effects. Stress is defined as a physiological response to demanding environmental or psychosocial conditions which can cause uncertainty about future situations or events. Normally, its induction better prepares the organism to overcome these challenges and to reachieve homeostasis. A balanced stress response is essential for the optimal functionality of these organisms but conditions which cause extreme acute or chronic stress can lead to various negative effects.

Stress exerts profound effects on brain structure and function, and the hippocampus is one of the primary neural targets of stress-induced remodeling. The impact of stress on AHN depends on its duration, intensity, and controllability. Acute stress may transiently suppress progenitor proliferation, whereas chronic or unpredictable stress paradigms consistently produce long-lasting reductions in AHN [167,168]. The underlying mechanisms involve sustained activation of the HPA axis, resulting in elevated glucocorticoid levels that impair neuronal proliferation, differentiation, and survival [169,170]. In addition, stress-induced inflammatory cytokines, oxidative stress, and decreased trophic signaling—particularly BDNF downregulation—further contribute to reduced neurogenesis [78,171].

Experimental models such as chronic restraint stress, chronic unpredictable mild stress (CUMS), and social defeat have consistently shown diminished cell proliferation and neuronal differentiation within the DG [163,172] (Figure 4). These reductions in AHN are accompanied by behavioral correlates of depression and anxiety, suggesting that decreased neurogenesis contributes to stress-related psychopathology [165]. Pharmacological or behavioral interventions that reduce HPA axis hyperactivity or increase trophic signaling (e.g., antidepressants, environmental enrichment, voluntary exercise) can partially restore AHN and alleviate these behavioral deficits [168].

The effects of stress are not uniform along the hippocampal dorsoventral axis. A growing body of evidence demonstrates that ventral hippocampal neurogenesis is particularly sensitive to stress exposure, whereas dorsal neurogenesis is relatively more resilient. Chronic stress paradigms preferentially suppress cell proliferation and reduce immature neuron numbers in the ventral DG [78,170]. Conversely, the dorsal hippocampus, which is primarily involved in spatial and cognitive functions, exhibits smaller or transient changes in proliferative activity following similar stress conditions [77]. This regional asymmetry is in line with functional specialization since the ventral hippocampus regulates emotional and stress-related responses, while the dorsal hippocampus supports cognitive processing and memory.

Moreover, recovery of ventral neurogenesis following chronic stress correlates closely with antidepressant efficacy and behavioral normalization, emphasizing its role as a key mediator of emotional resilience [98,158]. In contrast, dorsal neurogenesis appears to contribute more to the restoration of cognitive flexibility after stress cessation. Overall, stress primarily targets the ventral AHN, reflecting its integration into limbic–endocrine circuits. Future studies should aim to delineate the molecular determinants of this regional vulnerability and to clarify how interventions that selectively restore ventral AHN might enhance emotional resilience without perturbing cognitive function. It should be noted that because stress responses involve coordinated hormonal, autonomic, and behavioral components, changes in AHN observed under stress conditions are best understood as reflecting the integrated action of these overlapping physiological systems.

### 3.3. Diet and Nutritional Modifiers of AHN

The relationship between nutrition and brain health has been thoroughly studied and it is now widely accepted that a balanced diet throughout an organism’s lifetime is of great importance in order to achieve optimal brain function. The World Health Organization (WHO) considers a healthy diet to be a principal preventative factor for the development of cognitive decline and dementia [173]. To better understand the impact of nutrition on mental health, researchers have studied the molecular mechanisms involved. Evidence suggests that synaptic plasticity and hippocampal neurogenesis are significantly affected by both the amount and the type of food consumed [21]. Many of these dietary and hormonal factors exert region-specific influences along the hippocampal axis, a distinction further examined in the following section.

#### 3.3.1. Caloric Restriction & High-Fat Diet

The restriction of caloric intake without the induction of malnutrition, known as caloric restriction (CR), has been associated with multiple health benefits and extended longevity [174]. In the hippocampus, CR influences the adult neurogenesis, though results remain inconsistent across studies. Some authors report that CR increases the number of newborn cells in the DG [175,176], possibly through the enhanced release of BDNF [177]. Others, however, suggest that CR may decrease the neurogenic potential of neural progenitor cells [178,179,180]. Age appears to be an important moderating factor: younger animals may be more susceptible to CR-induced reductions in neurogenesis, whereas older individuals often show maintained or improved neurogenic capacity under CR conditions [178].

Staples et al. [180] provided important insights into the phase- and region-specific effects of CR along the hippocampal dorsoventral axis. Their findings indicate that CR enhances the proliferation of neural progenitors while simultaneously reducing the survival of newborn neurons, and that these effects are confined to the ventral DG. In contrast, the same study found a reduction in total granule cell number restricted to the dorsal DG, whereas the ventral subregion remained unaffected. This apparent bidirectional impact underlines the need to assess AHN not only by cell counts but also by the functional integration of newborn neurons into hippocampal circuits. Region-specific vulnerability suggests that dietary interventions may exert paradoxical effects on cognition versus mood regulation, thereby enhancing emotional adaptability via ventral neurogenesis while potentially impairing dorsal, cognition-related plasticity.

Diets rich in saturated fats represent the opposite metabolic condition and are established risk factors for obesity, coronary artery disease, and several cancers [173]. Mounting evidence indicates that high-fat diets (HFDs) also impair brain health, leading to cognitive deficits and accelerated neurodegenerative processes such as dementia and Alzheimer’s disease (AD) [181,182,183,184]. The neurobiological mechanisms underlying these effects involve significant disruption of hippocampal neurogenesis, with the ventral hippocampus showing greater susceptibility to diet-induced damage than other subregions [185,186,187]. This inhibition of AHN is believed to arise from neuroinflammatory processes, mediated by activation of M1-type microglia, which release pro-inflammatory cytokines such as interleukin-1 (IL-1), interleukin-6 (IL-6), and tumor necrosis factor-α (TNF-α) [186,187,188]. Microglia express Toll-like receptor 4 (TLR4), which normally recognizes bacterial lipopolysaccharides (LPSs), but can also be activated by oxidized low-density lipoproteins (LDLs) generated during lipid metabolism [189,190,191]. Activation of this pathway promotes microglial polarization toward the M1 pro-inflammatory phenotype, producing an inflammatory environment that is unfavorable to neurogenesis, leading to decreased proliferation and differentiation of newborn neurons [192].

Collectively, these findings reveal that caloric restriction and high-fat diet exert opposing effects on hippocampal neurogenesis, and that these effects are regionally distinct along the dorsoventral axis. While CR may differentially modulate proliferation and survival between dorsal and ventral hippocampus, HFDs predominantly compromise ventral neurogenic integrity through inflammatory mechanisms. Understanding how energy balance interacts with hippocampal subregional specialization will be critical for designing dietary interventions aimed at optimizing both cognitive and emotional brain health.

#### 3.3.2. Vitamins and Other Nutritional Factors

Vitamins and micronutrients are essential for neurodevelopment, neuronal maintenance, and neurotransmission, and their deficiencies are major contributors to central and peripheral nervous system disorders. Several vitamins play particularly important roles in AHN by regulating cell proliferation, differentiation, and survival.

B-complex vitamins, including vitamin B12 and folic acid, are critical for DNA synthesis and methylation, and therefore indispensable for NSC proliferation. Folic acid deficiency has been associated with reduced neurogenic rates—especially in the ventral hippocampus, where emotional and stress-regulatory circuits are more sensitive to metabolic disturbances [193]. Vitamin B1 (thiamine) deficiency, which causes Wernicke encephalopathy, dramatically decreases the survival of newborn hippocampal neurons [194]. This reduction likely contributes to the cognitive and memory deficits seen in Wernicke’s pathology, due to the strong interconnectivity between the mammillary bodies and hippocampus involved in episodic memory formation and confabulation [194,195].

Vitamin A (retinol), through its active metabolite retinoic acid (RA), functions as both an antioxidant and a regulator of gene expression. Vitamin A deficiency reduces hippocampal neurogenesis, likely through impaired cell proliferation, whereas RA supplementation restores it [196]. Conversely, chronic RA administration, as used in some dermatological treatments, has been reported to suppress AHN [197], underscoring the need for precise dose regulation.

Vitamin C (ascorbic acid), a potent antioxidant and enzymatic cofactor, enhances hippocampal neurogenesis and counteracts D-galactose-induced brain aging by promoting newborn neuron maturation [198]. Postnatal vitamin C deficiency reduces total hippocampal neuron counts in guinea pigs [199]. Overall, vitamin C appears to support neurogenesis not only in the hippocampus but also in other neurogenic niches (for review, see [200]).

Among trace elements, zinc is particularly important for neurogenic integrity. Zinc supplementation promotes survival of neuronal progenitor cells, likely through the inhibition of apoptosis [201]. Similarly, selenium supports AHN by increasing neural progenitor survival and reducing neuroinflammation. Its role is mediated via PI3K/Akt/Nrf2 and BDNF/TrkB signaling pathways [202]. Although regional differences have not yet been systematically explored, selenium’s neuroprotective role may extend preferentially to the ventral hippocampus, where oxidative and inflammatory stress are more prominent.

Omega-3 polyunsaturated fatty acids, including eicosapentaenoic acid (EPA) and docosahexaenoic acid (DHA), are essential for maintaining neuronal membrane fluidity and synaptic function. DHA supplementation increases hippocampal neurogenesis by enhancing newborn neuron survival [203], while long-term dietary EPA/DHA intake reverses age-related declines in neurogenesis [204]. These effects may be more pronounced in the ventral hippocampus, consistent with the antidepressant and anxiolytic effects of omega-3 fatty acids and the ventral region’s role in emotional regulation.

Flavonoids, a broad class of plant-derived polyphenols, have received considerable attention for their antioxidant, anti-inflammatory, and neuroprotective effects. They modulate oxidative stress and neuroinflammation and may act directly on neuronal progenitors to promote proliferation and differentiation (for review, see [205]). Curcumin, the yellow polyphenolic pigment found in Curcuma longa (turmeric), shares similar properties. It enhances AHN by rescuing newborn neurons from apoptosis and activating the Wnt/β-catenin pathway while increasing BDNF expression [206,207].

Collectively, these findings highlight the effects of vitamins, trace elements, and bioactive dietary compounds on AHN. While evidence for dorsoventral specificity remains limited, deficiencies in vitamins such as folate or nutrients affecting oxidative and inflammatory homeostasis appear to impair ventral neurogenesis preferentially, potentially leading to emotional dysregulation. In contrast, nutrients that enhance synaptic and trophic signaling (e.g., omega-3 fatty acids, vitamin A, curcumin) may also strengthen dorsal neurogenesis linked to cognitive resilience. Future work should address how specific micronutrient pathways interact with hippocampal subregional specialization to support both cognitive and emotional health. 

## 4. AHN from a Dorsoventral Perspective

This section integrates the modulatory influences reviewed above within the framework of dorsoventral hippocampal organization. Although the dorsoventral organization of the hippocampus is well characterized in rodents, corresponding distinctions along the human hippocampus are less clearly defined due to methodological limitations.

The hippocampus exhibits a remarkable functional and structural differentiation along its dorsoventral axis, which extends from the dorsal (septal) to the ventral (temporal) pole. This organization is accompanied by distinct, yet interrelated, behavioral and cognitive functions subserved by each region: the dorsal hippocampus is primarily engaged in spatial and cognitive processing, whereas the ventral hippocampus is preferentially involved in emotion, stress regulation, and affective behavior [208,209], which are accompanied by specializations at the molecular and cellular levels [209,210,211,212]. Correspondingly, AHN also displays regional heterogeneity in baseline activity, regulation, and functional relevance [158,213,214]. Table 1 summarizes the intrinsic, behavioral, and dietary modulators of AHN for which dorsoventral differences have been reported, providing an overview of the factors discussed in the following sections.

### 4.1. Dorsoventral Heterogeneity of Neurogenesis

Early studies suggested that cell proliferation and immature neuron density (marked by doublecortin expression) were higher in the dorsal DG of rodents, implying a predominance of neurogenesis related to cognitive functions [59,156]. However, subsequent research revealed that mature adult-born neurons are more numerous in the ventral DG, indicating that overall neuronal addition may actually favor the ventral region. This contradiction could be the result of increased newborn neuron survival in the ventral hippocampus and be dependent on species, age, and environmental conditions.

In marmosets, for example, a relatively homogeneous distribution of doublecortin (DCX)-positive neurons was observed along the septotemporal axis [214], whereas in rodents, dorsal enrichment in proliferation coexists with stronger neurogenic modulation in the ventral domain under specific physiological or stress conditions [134,215]. This apparent paradox suggests that regional differences in neuronal survival and maturation rates strongly influence overall neurogenic output.

In general, baseline levels of progenitor proliferation and immature neuron density are higher in the dorsal DG, and cognitive stimulation through environmental enrichment and learning paradigms tends to preferentially enhance dorsal neurogenesis [134,156] supporting its tonic role in information encoding, spatial learning, and pattern separation in the dorsal hippocampus [59,156]. In contrast, the ventral DG typically exhibits lower baseline neurogenesis, but it shows increased sensitivity to environmental, hormonal, and emotional stimuli. Chronic stress paradigms, such as social defeat and unpredictable chronic stress, preferentially suppress ventral neurogenesis, while antidepressant treatments predominantly restore neurogenesis in this same region. Neurogenesis in the ventral hippocampus is preferentially affected by stress, antidepressants, and neuromodulators [77,78,98] (O’Leary et al., 2014; Alvarez-Contino et al., 2023; Liu et al., 2023). For instance, chronic stress or elevated Tau expression selectively suppresses ventral neurogenesis and promotes anxiety-like behavior [78], whereas antidepressant treatments such as fluoxetine or GALR2/Y1R agonists restore neurogenesis primarily in the ventral DG and alleviate depressive phenotypes [98,216]. Thus, while the dorsal hippocampus maintains stable, tonic neurogenesis supporting cognitive operations, the ventral hippocampus is dynamically regulated by stress hormones, immune signals, and antidepressant treatments, reflecting its emotional and adaptive plasticity. This regional specialization aligns with the established functional dichotomy with the dorsal hippocampus involved mainly in spatial/cognitive processing and the ventral hippocampus more involved in emotional/stress regulation.

### 4.2. Molecular Mechanisms Underlying Dorsoventral Differences

The molecular mechanisms underlying dorsoventral differences in neurogenesis are increasingly understood. For instance, the miR-17-92 cluster, is more highly expressed in the ventral DG, promoting neurogenesis and exerting antidepressant-like effects [217]. Similarly, region-specific signaling via neuregulin-1 and β_2_-adrenergic or GABAergic pathways selectively promote ventral neurogenesis and emotional resilience [98,218,219,220]. Conversely, dorsal neurogenesis is preferentially modulated by learning and cognitive stimulation, with higher increases in the dorsal hippocampus following environmental enrichment, physical activity, and training tasks [134,156]. Transcription factors such as AP2γ or JNK1 influence proliferation and neuronal differentiation with predominant effects in the dorsal hippocampus, consistent with its cognitive specialization [220,221], reinforcing a distinction between cognitive-tonic and emotional-plastic neurogenic segmentation of the hippocampus. On the other hand, the miR-17-92 cluster shows higher expression in the ventral hippocampus and promotes neurogenesis with antidepressant-like effects [217,222]. In addition, JNK1 inhibition produces preferential enhancement of neurogenesis in the ventral hippocampus [220]. Also, the secreted frizzled-related protein 3 (sfrp3) exhibits a dorsoventral gradient that could potentially contribute to differential adult NSC activity between dorsal and ventral regions [223,224].

Norepinephrine and potassium chloride treatment significantly preferentially enhance neurosphere formation in temporal (ventral-equivalent) hippocampal progenitors regions compared with septal (dorsal-equivalent) ones, indicating higher intrinsic excitability and responsiveness of ventral neural stem cells [225,226].

Autophagy has also been recognized as an important cellular mechanism influencing adult hippocampal neurogenesis. Basal autophagic activity supports the maintenance of neural stem cell quiescence and the metabolic stability required for neuronal differentiation, while its impairment has been associated with reduced survival of newborn neurons and delayed maturation [227]. Several studies report that the enhancement of autophagy promotes progenitor proliferation and neuronal maturation, whereas autophagic deficits contribute to neurogenic impairments observed in chronic stress paradigms and in early phases of neurodegenerative disorders such as Alzheimer’s disease [228]. Although potential dorsoventral differences in autophagic regulation have not yet been systematically examined, the strong link between autophagy, cellular resilience, and stress responsiveness suggests that ventral DG neurogenesis may be particularly sensitive to autophagy-related modulation.

### 4.3. Functional Implications

Selective regulation of neurogenesis along this axis produces behaviorally dissociable outcomes, thereby supporting a dual neurogenic system. Neurogenesis in the dorsal hippocampus contributes to pattern separation, spatial learning, and memory precision, whereas ventral-born neurons integrate into circuits regulating affective and stress responses [157,158]. Suppression of ventral neurogenesis increases anxiety and stress susceptibility [78], while enhancement of ventral hippocampus neurogenesis confers stress resilience and antidepressant-like effects [158,216]. Importantly, neurogenic responses to environmental or pharmacological modulation, including exercise, enriched environments, or antidepressant drugs, show regionally distinct outcomes. Manipulations that potentiate dorsal neurogenesis correlates with cognitive flexibility and spatial memory performance [134,156]. Environmental enrichment or voluntary exercise can upregulate neurogenesis in both hippocampal segments but contributes differentially to behavioral improvement: dorsal neurogenesis supports cognitive recovery, whereas ventral neurogenesis appears to mediate emotional normalization [134]. Interestingly, adult-born neurons in both dorsal and ventral hippocampus contribute to contextual fear conditioning, but likely through distinct mechanisms. Dorsal adult-born neurons may contribute to the acquisition and recall of context representations, while ventral adult-born neurons may modulate the ability of these representations to activate fear via hippocampal projections to the amygdala [229,230,231,232]. These findings indicate that behavioral outcomes depend on the spatial locus of neurogenesis activation.

### 4.4. Integrative Perspective

Taken together, these findings support a bidirectional model in which dorsal and ventral neurogenesis function as complementary systems for adaptive behavior. The dorsal hippocampus functions as a stable, cognition-oriented generator with higher tonic neurogenic output, and the ventral hippocampus, characterized by lower baseline activity but greater modulatory plasticity, may function as a flexible regulator of affective behavior, supporting emotional adaptation to environmental and physiological challenges. As illustrated in Figure 5, modulatory influences exhibit distinct dorsoventral profiles, highlighting their predominant regional effects and directionality on AHN. This spatial specialization may have evolved to allow parallel yet integrated regulation of cognition and emotion via distinct neurogenic domains. Variability in dorsoventral neurogenesis patterns across species [214,233] and experimental models add further complexity to this framework, and suggest that region-specific susceptibility to stress, hormones, or disease factors could underlie differential neuropsychiatric vulnerabilities across species, thereby pointing to important clinical implications. For instance, understanding that ventral neurogenesis preferentially responds to antidepressant treatments while dorsal neurogenesis supports cognitive enhancement could inform precision medicine approaches for psychiatric and neurological conditions. The clinical implications of AHN will be further discussed in Section 5.

**Table 1 cells-15-00059-t001:** Summary of intrinsic, behavioral, and dietary factors that modulate adult hippocampal neurogenesis (AHN) and have been studied for potential dorsoventral differences.

Factor	Effect on Ventral AHN	Effect on Dorsal AHN	References
Learning	No change/Decrease	Increase	[156,234]
Chronic Unpredictable Stress	Decrease	No change	[169]
Early Life Stress	No change	Decrease	[171]
Physical Activity	No change/Increase	Increase	[41,90,133]
Prolonged Sleep Deprivation	Decrease	Decrease	[140]
Caloric Restriction	Increase/Decrease	Decrease	[180]
High-Fat Diet	Decrease	Decrease	[186]
Vitamin B1 (Thiamine)	No data	Increase	[194,235]
Flavonoids	No Change	Increase	[59]
Folic Acid	Increase	No change/Increase	[193]
Cortisol	Decrease ^1^	No change/Decrease ^1^	[39]
Estrogens	Increase/No change/ Decrease	Increase/Decrease ^2^	[53,59]
Androgens	Increase	Increase	[64]
Oxytocin	Increase	No change	[73]
Melatonin	Increase	Increase	[90]

^1^ Cortisol has a dose-dependent effect on AHN and could potentially increase it in low concentrations as explained in Section 3.1.1. ^2^ Estrogens have a sex-, time- and dose-dependent effect on AHN, as explained in Section 3.1.2.

## 5. Implications of Dorsoventral AHN Modulation in Neuropsychiatric and Neurodegenerative Disorders

A range of neuropsychiatric and neurodegenerative conditions involve hippocampal dysfunction, and accumulating evidence suggests that dorsoventral differences in AHN may contribute to their distinct cognitive and affective profiles. This observation led to the assumption that AHN may play a role in the pathogenesis and prognosis of these diseases with recent evidence further supporting this theory [236].

The dorsoventral differentiation of AHN provides a valuable anatomical and functional framework to interpret these alterations. Disorders with predominant cognitive impairments may involve dysfunction of dorsal neurogenesis, whereas those characterized by emotional dysregulation, anxiety, or motivational disturbances are more likely associated with ventral deficits. Reduced ventral neurogenesis is a recurring feature across mood and stress-related disorders, whereas symptoms associated with dorsal deficits are often prodromal in conditions marked by cognitive impairment such as AD.

### 5.1. Major Depressive Disorder & Anxiety Disorders

MDD is a psychiatric disorder classically characterized by persistent depressed mood and loss of interest in activities that were previously enjoyed by the diseased individual (anhedonia). When symptoms are severe, patients might experience suicidal ideation and/or resort to self-harm [237]. Anxiety disorders are either chronic or episodic/situational states characterized primarily by excessive fear, worry, or nervousness that is disproportionate to the situation and interferes with daily functioning. Evidence shows reduced AHN in models of MDD and anxiety, particularly within the ventral hippocampus, which integrates stress, serotonergic, and neuroendocrine inputs [78,158].

Multiple studies have been conducted to explore structural differences in patients with MDD with evidence indicating that the hippocampi of the diseased are usually characterized by reduced gray matter volume. This finding is positively correlated with the duration of the illness [238,239]. However, evidence is not sufficient to support causality.

In addition, acute and severe forms of MDD are associated with an elevated cortisol response [48] which is detrimental to newborn neuron proliferation. These findings indicate that reduced rates of AHN could explain a part of MDD pathogenesis [240]. This hypothesis is further supported by the fact that the same factors which have been found to be beneficial for the mood of MDD patients, such as healthy diet and lifestyle, are also shown to have positive effects in neurogenic activity. Moreover, antidepressants used for the treatment of MDD, such as Selective Serotonin Reuptake Inhibitors (SSRIs), Tricyclic Antidepressants (TCAs) and Monoamine oxidase Inhibitors (MAOIs) have been shown to significantly increase the rates of AHN through an increase in the number and maturation neural progenitor cells [216,241,242,243,244]. These neurogenic enhancements seem to in fact be essential for the positive effect of antidepressants, since mice with abolished neurogenic potential fail to show behavioral improvement following chronic antidepressant administration [245,246]. Chronic stress, glucocorticoid excess, and pro-inflammatory states suppress ventral proliferation and neuronal differentiation, effects that can be reversed by antidepressant treatment or by stimulation of neurotrophic signaling [77,98,216]. Antidepressants such as SSRIs, Agomelatine or GALR2/Y1R agonists preferentially restore neurogenesis in the ventral domain, paralleling the normalization of emotional behavior [216,247,248]. The neurobiological mechanism through which the neurogenic benefits of antidepressants are mediated is potentially related to the activation of the Wnt pathway in neural progenitor cells, induced by the increase in Serotonin levels and the subsequent activation of 5-hydroxytryptamine (5-HT) receptors and increased BDNF release [249,250]. The activation of the Wnt pathway through this mechanism has particularly been observed in the newborn neurons located in the Ventral Hippocampus [250]. Another proposed mechanism through which antidepressants and serotonin enhance AHN levels is through the increase in hippocampal BDNF levels [251,252].

However, whether enhanced neurogenesis is a cause or consequence of antidepressant efficacy remains unresolved. Longitudinal imaging or biomarker studies could clarify if AHN recovery precedes mood improvement, thereby testing the neurogenesis hypothesis of depression more directly.

Furthermore, it should be noted that Nonpharmacologic treatments of MDD, such as Electroconvulsive Therapy (ECT) and vagus nerve stimulation, have also been associated with significantly enhanced AHN [253].

From a more psychological and clinical perspective, a potential mechanism through which these neuronal deficits seen in the hippocampus of untreated MDD patients can lead to the expression of MDD symptomatology is the dysfunction of the existing and constantly diminishing circuitry to provide emotional variability to the patient. This mechanism could partly account for the persistent depressive mood and anhedonia observed in patients with MDD (Figure 6). The birth of new neurons could enhance the function of the ventral hippocampus and offer the patient an increase in emotional diversity apart from the persistent depressed mood MDD is characterized by. Additionally, researchers have shown that neuronal deficits in the dorsal hippocampus of MDD patients can also lead to a diminished memory performance [254]. This subpar performance can interfere with the patient’s daily life and can lead to their frustration and disappointment which can further negatively impact their mood. This cycle of events can confine MDD patients in a depressive state and psychiatric intervention at this stage should be deemed necessary (Figure 6).

It should be noted, however, that although decreased neurogenic rates can not solely lead to depressive behaviors [255], we propose that decreased neurogenic rates can diminish the ability of an individual to psychologically overcome daily psychosocial stressors and can therefore predispose to the development and unfavorable prognosis of MDD. In addition to depression, reduced AHN has also been observed in chronic anxiety models, consistent with the ventral hippocampus’ role in affective regulation [168]. While the evidence base is smaller, these findings suggest that impaired neurogenesis may contribute more broadly to stress-related psychopathology.

### 5.2. Schizophrenia

Schizophrenia is a chronic neuropsychiatric disorder, in which the patient experiences positive symptoms, such as hallucinations, delusions, disorganized speech and behavior, as well as negative symptoms, such as impaired cognitive performance, social isolation, reduced motivation and flat affect. These symptoms cause patients to face significant disability, impairing their ability to function in daily life and can reduce their life expectancy by approximately 13–15 years [256].

The brains of patients suffering from Schizophrenia have been well studied by researchers with results indicating that the hippocampi are among the key regions affected. More specifically, a meta-analysis comparing patients suffering from chronic schizophrenia and healthy controls showed that the average hippocampal volume was significantly smaller in schizophrenia patients [257]. In addition, altered AHN has also been described in schizophrenia, though evidence is more heterogeneous. Neurodevelopmental models and patient studies suggest reduced proliferation and impaired integration of new granule cells in the DG [253,258]. There have been reports of decreased expression of cell proliferation markers and impaired maturation of newborn neurons in the adult hippocampus of schizophrenia patients, indicating that the structural differences mentioned above could be attributed to low neurogenic potential [253]. Weissleder et al. have explored the pathological origin of these observations and suggested that an increase in immune cell and macrophage population could explain this neurogenic impairment [259]. It should be noted, however, that since Schizophrenia has strong heritability and usually presents at a young age, the genetic component of the disease cannot be dismissed. Many genes have been associated with an increased risk for schizophrenia development, with DISC1, Dgcr8, FEZ1, NPAS3, SNAP-25 and a-CaMKII being of great interest. Notably, all of these genes not only predispose to schizophrenia but are also significantly important in the neurogenesis, playing roles in NSC proliferation and newborn neuron maturation [253]. Moreover, a recent study showed that a mutation in the Neuregulin 1 Nuclear Signaling Pathway can cause a dysregulation of schizophrenia susceptibility genes and lead to significantly reduced synaptic plasticity NSC proliferation, even further supporting the existence of an association between Schizophrenia and Neurogenesis [260].

Cognitive and negative symptoms, including working-memory deficits and contextual disorganization, are likely associated with dorsal hippocampal dysfunction, given the dorsal domain’s role in cognitive mapping and mnemonic precision. Conversely, positive and affective symptoms, such as stress-sensitive hallucinations or emotional dysregulation, may reflect ventral hippocampal hyperactivity and aberrant coupling with limbic regions [261]. In rodent models, ventral hippocampal hyperexcitability drives dopaminergic dysregulation in the VTA and NAc, a hallmark of psychosis [262]. If adult-born ventral neurons normally function to modulate or buffer this excitatory output, impaired ventral AHN could exacerbate the limbic overdrive underlying positive symptoms. Conversely, deficient dorsal AHN could impair contextual binding and contribute to cognitive disorganization.

In summary, while Schizophrenia symptoms cannot solely be attributed to low neurogenic rates, the disease’s underlying pathophysiology could lead to impaired AHN which could explain some of the structural differences and the symptoms that the patients present with, especially the ones related to cognitive and affective functions. The genes that have been implicated to predispose individuals to schizophrenia, could play such an important role in the birth and maturation of neurons, that both the developing and developed brain are affected. A possibility that also cannot be excluded, however, is that these adult hippocampal neurogenic deficits observed in schizophrenia patients, could also be resultant from the emotional flattening and stress that these individuals experience. In addition, schizophrenia may involve a bidirectional imbalance along the dorsoventral axis of the hippocampus: ventral hippocampus overactivity with reduced inhibitory neurogenic modulation, and dorsal hippocampus under-recruitment impairing cognitive integration. Further research needs to be conducted to determine the exact role of the known predisposing genes in the neurogenesis and the precise role of dorsal and ventral hippocampus neurogenesis in the development of schizophrenia symptoms.

### 5.3. Alzheimer’s Disease

AD is a neurodegenerative disorder and the most common form of dementia, a condition characterized by cognitive decline which significantly impairs an individual’s ability to meet their basic needs independently. AD patients experience a progressive deterioration in memory performance, attention, reasoning ability, speech production and comprehension [263].

A characteristic finding in brains of AD patients is significant brain atrophy and neuronal death in many brain areas, including the hippocampal formation. This degeneration is thought to be caused by the impaired clearance and accumulation of senile plaques and neurofibrillary tangles in the affected brain tissue [264,265]. Strikingly, many of the non-genetic risk and protective factors for AD have also been associated with impairment and enhancement of AHN, respectively (Table 2). This observation has led many researchers to further explore the relationship between AHN and AD with evidence indicating that neurogenic potential is greatly diminished in AD patients compared to healthy adults [5]. Furthermore, other conditions that have been associated with low adult hippocampal neurogenic rates, such as MDD, have also been considered as risk factors for the development of AD [266].

Though the decline in AHN in AD is widespread, it is, however, particularly evident in the dorsal hippocampus, consistent with the dominance of cognitive and mnemonic impairments in the disease [267]. In addition, loss of progenitor proliferation and impaired neuronal differentiation parallel the progression of hippocampal atrophy and memory deficits. While both hippocampal segments eventually deteriorate, the selective vulnerability of the dorsal hippocampus corresponds to the early predominance of spatial and episodic memory decline, whereas the ventral hippocampus contribution may become more evident in later stages, when anxiety and affective disturbances emerge.

Collectively, the above findings suggest that altered AHN may contribute to AD vulnerability. Although reduced neurogenesis alone cannot account for AD pathogenesis, it may interact with other pathogenic processes, such as senile plaques (β-amyloid accumulation) and neurofibrillary tangles (tau pathology), to influence disease onset and progression. Factors detrimental to AHN, such as chronic stress or a high-fat diet, and factors that enhance it, such as physical exercise and learning, have also been associated with cognitive function in aging. Activities that promote dorsal neurogenesis in particular have been linked to improved mnemonic flexibility and to the maintenance of cognitive reserve, which is known to mitigate AD risk [268,269]. Table 2 summarizes AD-related risk and protective factors that have been reported to differentially modulate dorsal or ventral AHN, emphasizing potential mechanistic connections between dorsoventral specialization and AD susceptibility. Finally, the combined effects of neurotoxic molecule accumulation and environmentally induced reductions in AHN may act additively to reduce the aging brain’s resilience to neurodegenerative processes such as those seen in AD.

**Table 2 cells-15-00059-t002:** A summary of acquired factors discussed in this review and their impact on the development of Alzheimer’s Disease (AD) in comparison to their effect on AHN.

Factor	Protective/Risk Factor for AD	Effect on AHN	References
Learning/Education	Protective	Increase (Dorsal)	[149,268]
Physical Activity	Protective	Increase (Dorsal)	[41,270]
High-Fat diet	Risk Factor	Decrease	[185,187,271]
Sleep Deprivation	Risk Factor	Decrease	[140,272]
Stress/Depression	Risk Factor	Decrease (Ventral)	[167,168,240,273]

### 5.4. Addiction

Substance and Nonsubstance addiction are neuropsychiatric disorders in which patients have a strong and repeated desire to seek and consume a certain substance or perform certain activities. This desire is usually characterized by tolerance, with patients each time seeking a greater amount or an increased exposure to the substance or activity they are addicted to. The individuals who fail to satisfy this need can develop withdrawal symptoms, which makes abstinence from the addicting factor even more challenging [274].

The neurobiology of addiction involves the aberrant activation of the brain’s dopaminergic reward system, in which neurons from the midbrain’s VTA project to multiple areas including the frontal cortex, NAc and the hippocampus. The increased activation of Dopamine receptors, especially in the NAc, is implicated for the development of addiction’s symptomatology. The hippocampal formation also plays a vital role in the pathophysiology of addiction through efferent and afferent neuronal projections towards and from the VTA and NAc, and is responsible for the storing of contextual memories related to the addictive factor and possibly for initiating the driving force towards its acquirement [253,275,276]. In particular, addictive behaviors engage neural circuits linking the hippocampus to reward and motivation centers, particularly the NAc and VTA, which receive strong input from the ventral hippocampus. Ventral hippocampal projections provide contextual information that can bias reward seeking and drug craving [277].

Regarding substance addiction, the vast majority of addictive drugs, including Nicotine, Opioids, Cocaine, Amphetamine and 3,4-Methylenedioxymethamphetamine (MDMA), has been associated with either decreased NSC proliferation or decreased survival of newborn neurons in the hippocampi of rodents (for review, see [278]). The susceptibility to addiction has also been associated with AHN, as individuals suffering with other conditions that are considered to decrease neurogenic rates, such as MDD, Stress and Schizophrenia, are also more likely to become addiction victims [279,280,281]. This claim is further supported by the fact that ablation of AHN makes rat models more susceptible to cocaine addiction [282]. Notably, another study demonstrated that blocking ventral AHN levels led rats to preferring immediate small rewards over greater delayed ones, a finding consistent with addictive behavior [157]. Furthermore, activities that are known to increase neurogenic rates, such as physical exercise, have been found to be protective against relapse [283,284]. These observations led many researchers to believe that AHN plays a vital role in the prevention as well as treatment of addiction [285,286] and activities that further enhance it, such as physical activity, healthy diet, sufficient sleep and stress relief, could be advised to patients to further minimize their chances of relapse and succumbing to addiction in the first place. Interestingly, experimental disruption of AHN enhances cue-induced reinstatement of drug seeking, whereas restoration of neurogenesis reduces relapse susceptibility [285]. The ventral hippocampus, through its connectivity with limbic-reward circuits, may act as a contextual gate, linking environmental cues to motivational drive. Thus, ventral AHN could serve a homeostatic role, dampening hyper-reactivity of reward circuits and promoting behavioral flexibility. In contrast, dorsal AHN likely contributes to the cognitive control of reward-related learning and extinction, processes such as discriminating safe from drug-associated contexts or updating maladaptive memories. A dorsoventral imbalance, with ventral suppression and dorsal rigidity, may therefore underlie both emotional vulnerability and impaired cognitive regulation during addiction.

## 6. Discussion

The existence and functional significance of AHN in humans remain debated, yet accumulating evidence suggests it persists at low but functionally relevant levels across life. While early reports provided compelling evidence for ongoing neuronal birth in the adult hippocampus, subsequent studies have questioned its prevalence, proposing that neurogenesis may decline to negligible levels in adulthood. Recent findings, however, suggest a more nuanced picture: AHN may persist throughout life but at levels that vary with age, disease state, and methodological approach. In particular, studies using sensitive markers and rigorous tissue preservation consistently detect immature neurons even in aged individuals, whereas negative findings often arise from postmortem samples with compromised integrity. Beyond the presence vs. absence dichotomy, the critical question is whether the observed levels are functionally relevant. Evidence linking reduced AHN with MDD, AD, and stress vulnerability argues that even modest neurogenesis could shape plasticity in human circuits. Resolving this debate will require standardized methodologies and, crucially, the development of non-invasive tools to quantify neurogenic activity in the living human brain.

The growing interest in AHN and decades of research have led to significant progress in understanding the neurobiological mechanisms underlying this process. Additionally, notable studies have not only identified factors that influence AHN but have also examined which stages of neurogenesis they affect. Some studies have even explored whether these factors exert subregion-specific effects within the hippocampal formation (Table 1). This detail, however, is commonly overlooked, as most studies assess AHN levels in the entire hippocampus. Given the undeniable evidence that the dorsal/posterior and ventral/anterior hippocampus differ not only functionally but also electrophysiologically [210,212], we urge future research to present data separately for these two structures, treating them as distinct entities. From our review of the literature, we consider worth highlighting the increased sensitivity of the ventral hippocampus to glucocorticoids [39], which are closely associated with stress, and the heightened plasticity of the dorsal/posterior hippocampus in response to learning challenges [154,155,156]. Given the role of the ventral hippocampus in affective functions and the role of the dorsal hippocampus in cognitive functions, these findings suggest the possibility of a region-specific recruitment of newborn neurons by the hippocampal DG in accordance with the brain’s functional demands. From a cellular perspective, this recruitment could be mediated through an increased survival and maturation to apoptosis ratio of newborn immature neurons during the early survival phase of the neurogenesis in response to stimuli from the affective or cognitive challenges. This hypothesis is of great interest and should be further explored.

Furthermore, we demonstrated findings suggesting a negative correlation between the levels of AHN and the development of neuropsychiatric disease, such as MDD and AD. However, while current evidence does not establish causality, the possibility that low AHN levels may predispose individuals to these conditions and influence their prognosis warrants further investigation. Experiments have been conducted involving mice with evidence indicating that AHN enhancement can promote resilience to stress and cognitive impairment [287,288]. Therefore, modulation of AHN levels through utilization of the easily modifiable factors affecting them could be explored in clinical practice, as an attempt to conservatively enhance AHN rates may complement existing treatments, potentially improving prognosis and treatment efficacy for neuropsychiatric diseases. Moreover, given the importance of AHN levels in the prognosis and their potential in the treatment of mental disease, AHN rates have been proposed as a potential biomarker for mental health, although reliable non-invasive quantification methods are still lacking.

Future methods enabling the quantification of neurogenic activity in living humans may predict the development of neuropsychiatric diseases long before symptom onset, allowing physicians to make early interventions to delay disease progression and thus significantly improve the quality of the patient’s life in the long term. Bridging rodent and human data remains a key challenge due to species-specific differences in hippocampal architecture and behavioral correlates. Future work integrating imaging, stem-cell models, and computational approaches may clarify AHN’s translational relevance.

Collectively, research on AHN in humans is still in its early stages. Skepticism regarding its functional and clinical significance, along with challenges in studying its levels in living humans, has discouraged further exploration. Yet, many questions remain unresolved. Could the stress resilience, which is linked to neurogenic enhancement in rodents, also apply to humans? Can the benefits of activities like physical exercise in alleviating neuropsychiatric symptoms be attributed to an enhancement of AHN rates? Could the utilization of modifiable factors that increase AHN rates in animals lead to clinical improvements or slow the progression of neurodegenerative diseases in humans? Addressing these questions could reignite scientific interest in AHN and unlock its clinical potential in preventing, diagnosing, and treating neuropsychiatric disease, while also taking us one step closer in understanding the neurobiology behind emotion, cognition and brain function as a whole. Moreover, recognizing the region-specific plasticity of AHN may ultimately refine our understanding of cognition–emotion interactions and inform region-targeted interventions for neuropsychiatric disease.

## Figures and Tables

**Figure 1 cells-15-00059-f001:**
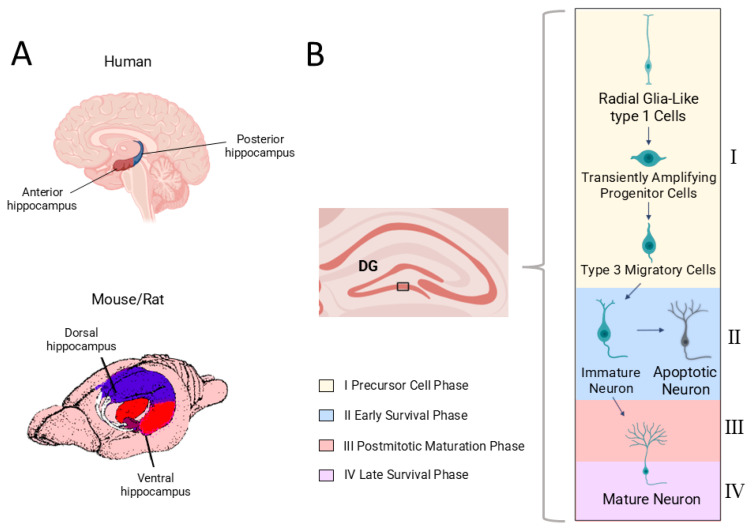
Overview of the adult hippocampal neurogenesis. (**A**). Anatomical comparison of hippocampal orientation in rodents (dorsal–ventral) and humans (posterior–anterior). (**B**). Schematic representation of the developmental progression of adult-born granule neurons, including precursor, early survival, maturation, and late survival phases. The developmental stages illustrated here are based on rodent studies, in which the cellular sequence of adult hippocampal neurogenesis has been experimentally defined and corresponds to the six-step trajectory described in the text. Corresponding stages in humans remain less well characterized. The human brain outline is included to provide anatomical context and facilitate comparison of hippocampal position between species.

**Figure 2 cells-15-00059-f002:**
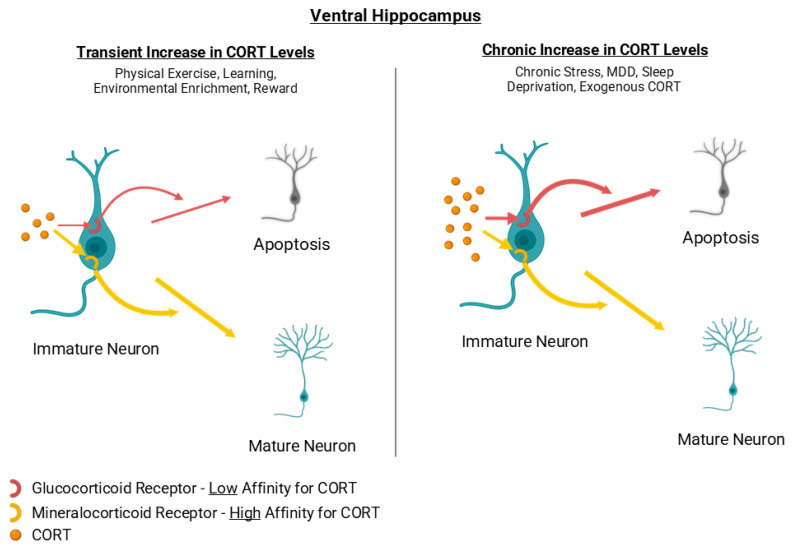
Glucocorticoid modulation of AHN. This schematic illustrates the effects of CORT on AHN via MR and GR signaling in the ventral hippocampus, shown here as a representative example of hormonal regulation. CORT is the most extensively studied endocrine modulator of AHN and provides well-established evidence for dorsoventral differences in hippocampal sensitivity. The ventral DG is highlighted due to its greater vulnerability to glucocorticoid-induced suppression compared with the dorsal DG. Transient CORT elevations (e.g., during exercise or enrichment) primarily engage MRs, which have high affinity for CORT and support neurogenesis when briefly activated. In contrast, chronic or sustained CORT elevations (e.g., during persistent stress or exogenous administration) saturate MRs and impair AHN via GR overactivation that leads to reduced proliferation, and impaired survival of newborn neurons. The figure illustrates the mechanistic principles specific to CORT that are discussed in Section 3.1.1.

**Figure 3 cells-15-00059-f003:**
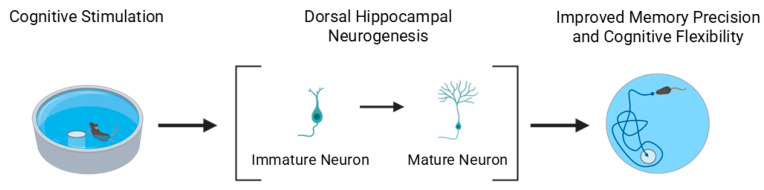
Functional contributions of immature and mature granule neurons to hippocampus. This schematic focuses on the two phases most relevant to the modulatory effects discussed in Section 3: the immature neuron stage, characterized by heightened excitability, enhanced synaptic plasticity, and sensitivity to environmental and hormonal influences, and the mature neuron stage, in which stable synaptic integration and circuit-specific functions predominate. By contrasting these two stages, the figure emphasizes the functional distinctions that underlie their differential responsiveness to modulatory factors. It should be noted that learning paradigms engage multiple interacting processes, such as attention, emotional arousal, reward, and stress signaling, so the observed effects on AHN probably reflect the combined influence of these factors rather than learning alone.

**Figure 4 cells-15-00059-f004:**
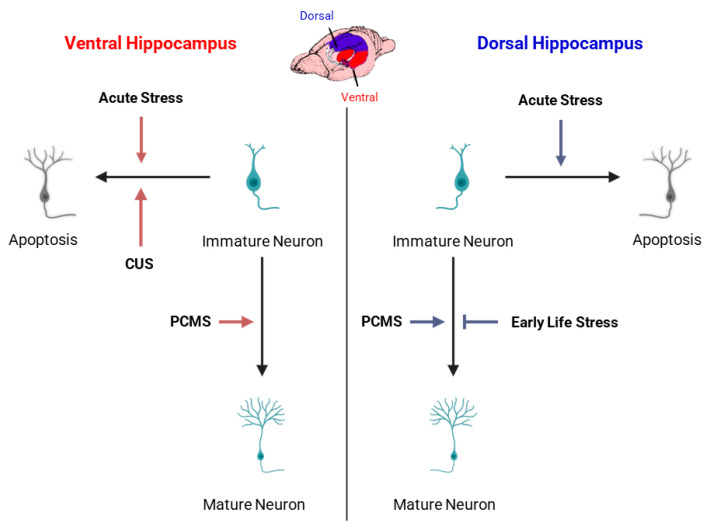
Effects of different types of stress on AHN from a dorsoventral perspective. Acute stress reduces neuronal survival in both subregions, chronic unpredictable stress (CUS) preferentially decreases ventral AHN, whereas predictable chronic mild stress (PCMS) can enhance neurogenesis and dendritic growth. Early life stress particularly impairs dorsal AHN. CUS: Chronic Unpredictable Stress; PCMS: Predictable Chronic Mild Stress.

**Figure 5 cells-15-00059-f005:**
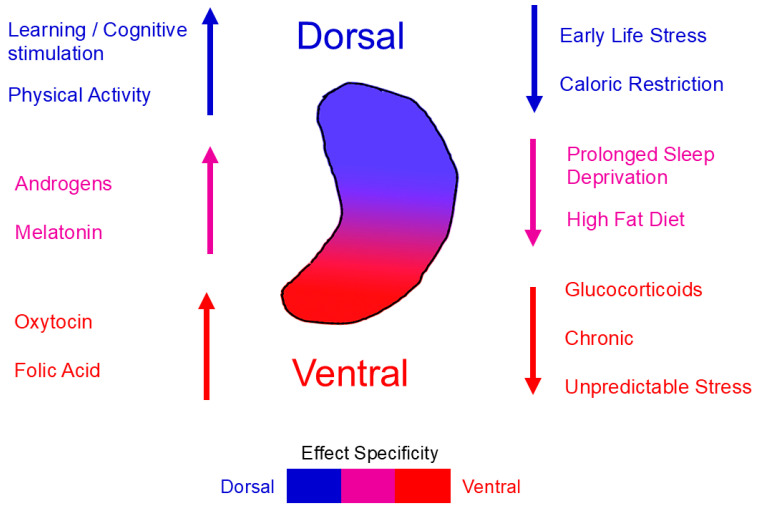
Dorsoventral specificity of modulatory factors influencing AHN. The schematic hippocampal outline illustrates the dorsoventral axis, color-coded from dorsal (blue) to ventral (red) regions. Upward and downward arrows indicate predominant enhancement or suppression of AHN, respectively, with color intensity reflecting regional specificity. Cognitive stimulation, physical activity, and certain hormones (e.g., androgens, melatonin) preferentially modulate dorsal or intermediate domains, whereas stress-related and metabolic challenges (e.g., glucocorticoids, chronic stress, high-fat diet) mainly impact ventral neurogenesis. This organization underscores the functional segregation of the hippocampus in mediating cognitive versus affective adaptations. Understanding these region-specific dynamics provides a neuroanatomical framework to interpret the selective vulnerability of AHN in neuropsychiatric disorders, discussed in Section 5.

**Figure 6 cells-15-00059-f006:**
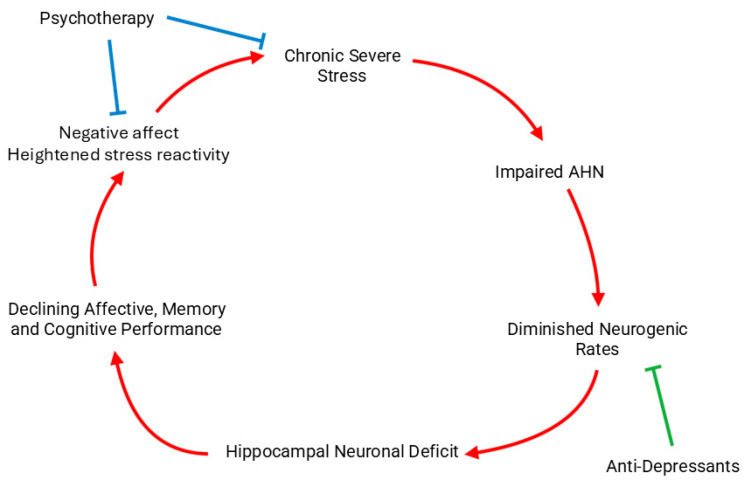
Cyclic interactions between chronic stress, impaired AHN, and depressive symptomatology. This schematic illustrates how chronic severe stress leads to impaired AHN and diminished neurogenic rates, which in turn contribute to hippocampal neuronal deficits and declining affective, cognitive, and memory performance, features commonly associated with MDD. These deficits may further exacerbate stress sensitivity and negative emotional states, creating a self-reinforcing cycle. Pharmacological antidepressants and psychotherapy are shown as interventions that disrupt this cycle by restoring neurogenic potential or reducing stress load. Although this figure does not depict dorsoventral distinctions, it serves to contextualize the functional consequences of AHN alterations within MDD and complements Figure 4 and Figure 5, which focus specifically on region-dependent hippocampal mechanisms.

## Data Availability

Not applicable.

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
