# Peer review of "Modifying Factors of Adult Hippocampal Neurogenesis: A Dorsoventral Perspective in Health and Disease"

_cells, 2025, doi:10.3390/cells15010059_

Round 1
Reviewer 1 Report
Comments and Suggestions for Authors
Cells - Review
Modifying Factors of Adult Hippocampal Neurogenesis: A Dorsoventral Perspective in Health and Disease
Ioannis Erginousakis and Costas Papatheodoropoulos
The dentate gyrus (DG) of the hippocampus is one of the few brain regions where adult neurogenesis (adN) continues throughout life. As part of the limbic system, the hippocampus is crucial for learning and memory, while ongoing neuron generation may help adapt neural circuits to environmental demands. In the DG’s subgranular zone (SGZ), neural stem cells give rise to new neurons in a regulated microenvironment shaped by glial cells, vasculature, and local signaling cues. The current review manuscript is excessively long, providing detailed descriptions of numerous hormones, vitamins, erythropoietin, and other factors that may indirectly influence or shape the neurogenic niche. It covers a wide range of topics, from diet, sleep, and stress to environmental influences and psychiatric disorders, thereby losing focus on the central theme of the dorsoventral axis. The manuscript contains several scientific flaws, and both organizational and conceptual clarifications are required to enhance its focus and readability. Please find my major comments and minor concerns below:
Major concerns and comments:
- The manuscript is excessively long and covers too many subtopics. The authors should narrow their focus, perhaps beginning with what distinguishes the dorsal and ventral HC, and if relevant, highlight differences between humans vs rodents.
- The author frequently cite review articles rather than original research papers, which undermines the scientific rigor of the manuscript.
- The few Figures included are overly simplified and lack clear rational. It is not evident why the authors selected only certain modulators or why they sometimes differentiate between dorsal and ventral and other times not. Similarly, why inclusion of immature, mature, and apoptotic neurons appear should be clarified.
- Only in Section 4.1. does the manuscript begin to address its main topic. Earlier sections should be constructed to align more closely with the central focus.
- Section 5 introduces an entirely new topic with a misleading headline. AD and addiction are not psychiatric disorders. The authors should either focus the review solely on these conditions given the involvement of the hippocampus, or limit the review to the other modulators discussed earlier.
Detailed comments
- line 35, neurons do not ‘multiply’ or ‘divide’! It is stem/progenitor cells that proliferate and eventually differentiate ionto neuron; similarily, in line 36, the phrase ‘neuronal proliferation’, is incorrect: neurons themselves do not divide
- line 36: the prominent seminal work demonstrating adult neurogeneiss was conducetd by ‘Altman and Das’ 1965, and should be cited as such
- line 38: The REF Bayer et al. 1982 [2] talkes about the general increase of the granule cell population
- lines 43-44: The transition from discussing humans back to general mammalian regions (primarily rodents) is confusing; clarify the distinction betw. species and specify when referring to rodent HC
- line 49, newly generated cells migrate and integrate functionally within the granule cell layer, not the SGZ
- line 55, the dorsoventral distinction is only introduced in Section 4, despite being the main focus of the review
- line 58, the phrase ‘reshape neural plasticity’ is awkward; consider using ‘modulate’
- Section 2 (line 75), consider clarifying the headline that this section refers to human data
- lines 103-125, the discussion of the 14C method is overly detailed and not central to the review focus; it also discusses rodent data which better fit earlier; the phrasing should be revised: adult N is ‘not mediated’ by stem cells, rather stem/progenitor cells devide and differentiate into neurons given specific environmental conditions; the process ‘comprises six developmental steps’ (see Kempermann et al 2004); i.e., the authors cite only a review by Kempermann, Song, and Gage instead of the original studies from these and other groups (e.g., Kronenberg et al. 2003, Steiner et al. 2006)
- Section 2, does not describe any mechanisms and instead only lists marker expression patterns observed in rodents
- Figure 1, the process illustrated is not established in humans; the figure should be revised reflecting the steps mentioned above for rodents
- Section 3., 3.1., REFs are missing
- Section 3.1.1, line 143 onward, this needs better organization, clarify whether data refer to human or rodent studies; begin with key findings and avoid mixing topics lines 162-170 are confusing and the distinction between acute and chronic should be defined (171)
- Section 3.1.6 (Melatonin, line 314), it should be noted that C57BL/6 mice do not produce melatonin
- Figure 2, the rational for including only one hormone and depicting it ventrally is unclear
- Section 3.2.1 Physical activity line 394, 3.2.2 Sleep line 432, 3.2.3 Learning line 477, these topics are not typically categorized as ‘behavior’ in research; human and rodent data are mixed again w/o distinction and key REFs are missing (e.g., van Praag et al., 1999, Klempin et al., 2013)
- Figure 3, the rational is unclear; why are only immature and mature neurons shown, w/o including the discussed developmental steps
- 2.4 Reward and 3.2.5 Stress, the rational for those are also not clear
- Tables, it is inconsistent to provide a table only for vitamins. Table 2 should be placed earlier in the manuscript if it summarizes all modulators discussed.
- Figure 6, this figure does not address the dorsoventral distinction in the hippocampus and seems unrelated to the main focus.
- from reference 34 onward, only author initials are provided
Author Response
Reviewer 1
Modifying Factors of Adult Hippocampal Neurogenesis: A Dorsoventral Perspective in Health and Disease
Ioannis Erginousakis and Costas Papatheodoropoulos
The dentate gyrus (DG) of the hippocampus is one of the few brain regions where adult neurogenesis (adN) continues throughout life. As part of the limbic system, the hippocampus is crucial for learning and memory, while ongoing neuron generation may help adapt neural circuits to environmental demands. In the DG’s subgranular zone (SGZ), neural stem cells give rise to new neurons in a regulated microenvironment shaped by glial cells, vasculature, and local signaling cues. The current review manuscript is excessively long, providing detailed descriptions of numerous hormones, vitamins, erythropoietin, and other factors that may indirectly influence or shape the neurogenic niche. It covers a wide range of topics, from diet, sleep, and stress to environmental influences and psychiatric disorders, thereby losing focus on the central theme of the dorsoventral axis. The manuscript contains several scientific flaws, and both organizational and conceptual clarifications are required to enhance its focus and readability. Please find my major comments and minor concerns below:
We thank the reviewer for carefully evaluating our manuscript and for providing a series of thoughtful and detailed comments. The reviewer’s constructive critique was extremely helpful in guiding substantial revisions, and we have carefully addressed each point below. We believe that these changes have significantly improved the clarity, accuracy, and overall quality of the manuscript. Changes in the revised manuscript are marked as follows: deleted text appears in red with strikethrough, and new or revised text is shown in red.
Major concerns and comments:
- I believe that a review on this specific topic (despite the availability of several strong existing reviews) should ideally be authored by experts deeply engaged in this field. This does not appear to be the case here.
Response: We thank the reviewer for this comment. We fully appreciate the importance of expertise when writing a review on a complex topic such as adult hippocampal neurogenesis. Our scholarly background lies in hippocampal physiology, dorsoventral functional specialization, and neural plasticity, and this perspective motivated us to synthesize and reinterpret existing neurogenesis research through the lens of hippocampal longitudinal differentiation. Although we have not contributed experimentally to all aspects of AHN, we believe that our disciplinary proximity allows us to offer a complementary and integrative view that may be useful to researchers in both fields. We also carefully revised the manuscript in light of the reviewer’s valuable suggestions, which helped us strengthen the focus, clarity, and scientific rigor of the review.
- The manuscript is excessively long and covers too many subtopics. The authors should narrow their focus, perhaps beginning with what distinguishes the dorsal and ventral HC, and if relevant, highlight differences between humans vs rodents.
Response: We thank the reviewer for this helpful observation. In the revised version, we have tried to improve focus and clarity. First, we substantially shortened portions of the Introduction (Section 1) by removing redundant background information, and Section 2 that contained historical overview, general background information, and the description of human AHN detection methods.
We also added a new paragraph early in the Introduction outlining the dorsoventral specialization of the hippocampus, as suggested. In addition, we inserted brief framing statements at the beginning of major sections to clarify the structure and to emphasize that the review concentrates on modulatory influences most relevant to dorsoventral differences. Where appropriate, we also highlight distinctions between human and rodent findings. Collectively, we believe that these revisions reduce the length and reorganize the manuscript, producing a clearer and more focused narrative that centers on dorsoventral differences, as recommended.
- The author frequently cite review articles rather than original research papers, which undermines the scientific rigor of the manuscript.
Response: We thank the reviewer for this important suggestion which was very helpful in improving the manuscript. We have now revised the manuscript to prioritize primary research studies over secondary review articles whenever original data is available. Specifically, we incorporated key foundational studies on AHN discovery, hormonal modulation, behavioral influences, and dorsoventral differences (including Altman & Das 1965; Kronenberg et al. 2003; Klempin et al. 2013, among others). Several review citations were replaced with the corresponding original studies, and additional original works were added to support mechanistic statements throughout the manuscript.
- The few Figures included are overly simplified and lack clear rational. It is not evident why the authors selected only certain modulators or why they sometimes differentiate between dorsal and ventral and other times not. Similarly, why inclusion of immature, mature, and apoptotic neurons appear should be clarified.
Response: We thank the reviewer for these comments. (a) We selected modulators based on the extent and robustness of available experimental evidence, focusing on factors that have been most widely studied in animal models and humans. For several modulators (e.g., stress, physical activity, hormones), sufficient data exist to evaluate differential effects along the dorsoventral axis, and we present these distinctions explicitly. For other modulatory factors, dorsoventral comparisons have not been systematically investigated, and in these cases we summarize their global effects on AHN while noting the absence of subregional data. We have now clarified this selection rationale in the manuscript to avoid any ambiguity regarding why some modulators include dorsoventral analysis and others do not.
(b) Regarding the figures, our intention was to provide schematic illustrations to aid conceptual understanding rather than detailed graphical representations of all underlying mechanisms, which could potentially tire the reader and obscure the main concepts. Nevertheless, following the reviewer’s suggestion, we have revised the figure legends to make the rationale and intended message of each figure more explicit, and we have clarified in the main text how each figure relates to the dorsoventral theme of the review. We hope these adjustments improve the clarity and perceived usefulness of the figures.
(c) The inclusion of immature, mature, and apoptotic neurons reflects the fact that AHN is a multistage process involving proliferation, survival, differentiation, synaptic integration, and long-term maintenance of newborn neurons. Different modulatory factors act at distinct stages of this trajectory. For example, stress and glucocorticoids primarily influence early survival, while learning and physical activity preferentially affect maturation and integration. Representing these cellular stages is therefore essential to illustrate how specific factors modify AHN quantitatively and qualitatively. We have now clarified this point in the revised text (Section 2, last paragraph).
- Only in Section 4.1. does the manuscript begin to address its main topic. Earlier sections should be constructed to align more closely with the central focus.
Response: We thank the reviewer for this insightful comment. To address this concern, we have revised the earlier sections of the manuscript to better align the manuscript with its central theme. Specifically, we added a paragraph early in the Introduction drawing the dorsoventral specialization of the hippocampus and explaining why this longitudinal organizational is essential for interpreting modulatory influences on AHN. We also removed redundant background material and inserted brief sentences at the beginning of relevant sections to emphasize their connection to the dorsoventral context. We believe that these revisions help to better define the focus of the review, and create a clearer conceptual transition towards the detailed dorsoventral analyses presented in Section 4.
- Section 5 introduces an entirely new topic with a misleading headline. AD and addiction are not psychiatric disorders. The authors should either focus the review solely on these conditions given the involvement of the hippocampus, or limit the review to the other modulators discussed earlier.
Response: We appreciate the reviewer’s point regarding the terminology and title of Section 5. In the revised manuscript, we have modified the section title to “Neuropsychiatric and Neurodegenerative Implications of Dorsoventral AHN Modulation” to avoid the misleading impression that all of the conditions discussed fall exclusively within the category of neuropsychiatric disorders. We believe that this title more accurately reflects the inclusion of Alzheimer’s disease and other neurodegenerative conditions alongside mood and stress-related disorders. Furthermore, we have strengthened the transition into Section 5 by clarifying earlier in the Introduction that alterations in AHN have been implicated across a spectrum of disorders that span both neuropsychiatric and neurodegenerative domains. We have also revised the introductory paragraph of Section 5 to explicitly state the rationale for examining these conditions together, namely, their involvement of hippocampal circuits and the evidence that dorsoventral neurogenic alterations may contribute to their distinct symptom profiles. We hope these changes resolve the relevant concern and improve the clarity and coherence of the manuscript.
Detailed comments
- line 35, neurons do not ‘multiply’ or ‘divide’! It is stem/progenitor cells that proliferate and eventually differentiate ionto neuron; similarily, in line 36, the phrase ‘neuronal proliferation’, is incorrect: neurons themselves do not divide
Response: We thank the reviewer for this important observation. In the revised manuscript, the sentences referring to neurons “multiplying/dividing” have been removed, and all terminology has been corrected to refer appropriately to the proliferation of neural stem and progenitor cells. We have reviewed the manuscript to ensure that no inaccurate expressions of neuronal division remain. We appreciate the reviewer’s attention to this detail.
- line 36: the prominent seminal work demonstrating adult neurogeneiss was conducetd by ‘Altman and Das’ 1965, and should be cited as such
Response: We thank the reviewer for this clarification. We have revised the text to explicitly cite the seminal 1965 study by Altman and Das at the point where the first evidence for adult neurogenesis is discussed. This correction appropriately acknowledges their foundational contribution to the field.
- line 38: The REF Bayer et al. 1982 [2] talkes about the general increase of the granule cell population
Response: We appreciate the reviewer’s attention to the accuracy of historical references. In the revised manuscript, the sentence referring to the Bayer et al. (1982) study has been removed as part of our effort to reorganize the historical overview and improve precision. The remaining text now cites only the key primary studies directly related to adult neurogenesis (Altman 1962 & Altman and Das, 1965).
- lines 43-44: The transition from discussing humans back to general mammalian regions (primarily rodents) is confusing; clarify the distinction betw. species and specify when referring to rodent HC
Response: We thank the reviewer for pointing out this ambiguity. We have revised the relevant sentences to clarify when we are referring specifically to rodent data. In particular, we now state explicitly that quantitative and mechanistic descriptions of AHN derive primarily from rodent studies. We also added brief species markers at the beginning of the dorsoventral paragraph to avoid confusion when going from human findings to rodent evidence.
- line 49, newly generated cells migrate and integrate functionally within the granule cell layer, not the SGZ
Response: We thank the reviewer for making us aware for this inadvertent mistake. We have revised the sentence accordingly to indicate that proliferation occurs in the SGZ, whereas newborn granule neurons migrate into the granule cell layer where functional integration takes place.
- line 55, the dorsoventral distinction is only introduced in Section 4, despite being the main focus of the review
Response: We thank the reviewer for highlighting this organizational issue. In the revised manuscript, we have introduced the dorsoventral specialization of the hippocampus early in the Introduction to establish the central theme from the outset. This new paragraph outlines the functional significance of the dorsoventral axis and explains why it is essential for interpreting modulatory influences on AHN. We also added brief sentences in Sections 2 and 3 to better strengthen the connection to the dorsoventral context.
- line 58, the phrase ‘reshape neural plasticity’ is awkward; consider using ‘modulate’
Response: We agree that the phrase “reshape neural plasticity” is awkward, and we have replaced it with “modulate.”
- Section 2 (line 75), consider clarifying the headline that this section refers to human data
Response: At Section 2 we revised the heading to explicitly state that the content refers to human data: “Adult Hippocampal Neurogenesis in Humans”.
- lines 103-125, the discussion of the 14C method is overly detailed and not central to the review focus; it also discusses rodent data which better fit earlier; the phrasing should be revised: adult N is ‘not mediated’ by stem cells, rather stem/progenitor cells devide and differentiate into neurons given specific environmental conditions; the process ‘comprises six developmental steps’ (see Kempermann et al 2004); i.e., the authors cite only a review by Kempermann, Song, and Gage instead of the original studies from these and other groups (e.g., Kronenberg et al. 2003, Steiner et al. 2006)
Response: We thank the reviewer for these valuable suggestions. In the revised manuscript, we substantially shortened the description of the ¹⁴C birth-dating method to focus only on its conceptual relevance, and we removed details that were not central to the aims of the review. We also clarified the phrasing to indicate correctly that adult neurogenesis is mediated by the proliferation and differentiation of neural stem and progenitor cells under specific environmental conditions. In accordance with the reviewer’s recommendation, we now describe the developmental trajectory of newborn neurons using the six-step framework outlined by Kempermann and colleagues (2004) and have added key primary studies that established these stages (e.g., Kronenberg et al., 2003; Steiner et al., 2006).
- Section 2, does not describe any mechanisms and instead only lists marker expression patterns observed in rodents
Response: We thank the reviewer for this helpful comment. In the revised manuscript, we have clarified the purpose of Section 2 and added a brief description of the mechanistic sequence underlying adult neurogenesis, including the activation of quiescent stem cells, proliferative expansion of progenitors, neuronal differentiation, migration into the granule cell layer, synaptic maturation, and long-term survival. We also reduced the emphasis on marker lists and revised the section to reflect more clearly that these markers are used experimentally to identify cells at different mechanistic stages of the neurogenic trajectory.
Also, in response to the reviewers’ suggestions for greater mechanistic clarity, we added a concise note on the role of autophagy, a cellular pathway increasingly recognized as relevant for neural stem cell maintenance, stress vulnerability, and neurodegenerative risk (Section 4.2. Molecular Mechanisms Underlying Dorsoventral Differences). This brief addition directly supports and enhances the existing discussion of factors that differentially influence ventral hippocampal neurogenesis.
- Figure 1, the process illustrated is not established in humans; the figure should be revised reflecting the steps mentioned above for rodents
Response: We thank the reviewer for this important clarification. We have revised the legend of figure 1 to state explicitly that the illustrated developmental trajectory of adult hippocampal neurogenesis reflects the sequence established in rodents, where these stages have been experimentally characterized. Evidence for corresponding stages in humans remains limited and indirect. We have retained the human brain scheme in the figure to provide anatomical context and to facilitate comparison of hippocampal location and orientation between humans and rodents. We hope these clarifications ensure appropriate interpretation of the figure.
- Section 3., 3.1., REFs are missing
Response: We thank the reviewer for noting this. We have carefully reviewed Section 3 and subsection 3.1 and added the appropriate references where they were missing. In particular, we now cite key primary studies demonstrating hormonal modulation of adult hippocampal neurogenesis (for glucocorticoids, estrogens, androgens, oxytocin, thyroid hormones, melatonin, GH/IGF-1, and erythropoietin).
- Section 3.1.1, line 143 onward, this needs better organization, clarify whether data refer to human or rodent studies; begin with key findings and avoid mixing topics lines 162-170 are confusing and the distinction between acute and chronic should be defined (171)
Response: We thank the reviewer for these constructive suggestions. In the revised manuscript, Section 3.1.1 has been reorganized to present glucocorticoid effects on AHN in a clearer sequence. We now begin with key findings from rodent studies, followed by mechanistic explanations involving MR/GR signaling, and then discuss the influence of acute versus chronic corticosterone exposure with explicit definitions. We also clarified the distinction between acute and chronic corticosterone exposure, reorganized the discussion of transient versus sustained CORT elevations, and removed or rephrased passages that mixed topics or created ambiguity. We believe that these revisions improve the structure, clarity, and interpretability of this subsection.
- Section 3.1.6 (Melatonin, line 314), it should be noted that C57BL/6 mice do not produce melatonin
Response: We are grateful to the reviewer for this important clarification. We have added a sentence in Section 3.1.6 noting that commonly used laboratory strains such as C57BL/6 mice have very low endogenous melatonin production due to a mutation in the enzyme arylalkylamine N-acetyltransferase. This clarification is relevant for interpreting melatonin-related findings in rodent studies and has now been incorporated into the text.
- Figure 2, the rational for including only one hormone and depicting it ventrally is unclear
Response: We thank the reviewer for this helpful observation. Figure 2 has now been clarified in the legend and main text. This figure is intended to illustrate the mechanistic basis of glucocorticoid modulation of AHN, using corticosterone as a representative example because it is the most extensively studied hormonal regulator of neurogenesis and provides the clearest evidence for dorsoventral differences. The ventral depiction reflects the well-documented greater sensitivity of the ventral DG to glucocorticoid-induced suppression, not an exclusion of dorsal involvement. We have revised the legend accordingly to make this rationale explicit.
- Section 3.2.1 Physical activity line 394, 3.2.2 Sleep line 432, 3.2.3 Learning line 477, these topics are not typically categorized as ‘behavior’ in research; human and rodent data are mixed again w/o distinction and key REFs are missing (e.g., van Praag et al., 1999, Klempin et al., 2013)
Response: We thank the reviewer for this valuable comment. In the revised manuscript, we renamed Section 3.2 from “Behavior” to “Lifestyle and Environmental Factors” to reflect more accurate categorization of the included topics. We also clarified the distinction between human and rodent evidence throughout subsections 3.2.1 - 3.2.3 by explicitly indicating when findings derive from rodent studies and when human data are available. In addition, we have added key primary references, including the seminal study by van Praag et al. (1999) demonstrating exercise-induced enhancement of AHN, and Klempin et al. (2013) showing region-specific activation of neurogenesis during running.
- Figure 3, the rational is unclear; why are only immature and mature neurons shown, w/o including the discussed developmental steps
Response: We thank the reviewer for this observation. Figure 3 has been clarified in the legend to indicate that it is intended to illustrate the functional distinction between immature and mature granule neurons in the dentate gyrus, rather than to depict the full developmental trajectory of adult-born neurons. The figure focuses on these two stages because they represent the phases most relevant to the modulatory effects discussed in Section 3 (e.g., activity-dependent survival, integration, and functional contributions). The earlier developmental steps are described in detail in figure 1 and in Section 2. We have revised the legend accordingly to make this rationale explicit.
- 2.4 Reward and 3.2.5 Stress, the rational for those are also not clear
Response: We thank the reviewer for this helpful comment. In the revised manuscript, we clarified the rationale for including Reward (Section 2.4) and Stress (Section 3.2.5) as distinct modulatory factors of AHN. Reward-related stimuli and stress represent two major behavioral and physiological states that exert robust and often opposing influences on hippocampal neurogenesis through well-defined neuroendocrine and neuromodulatory pathways. We have added introductory sentences to both subsections to explain their relevance and to specify when findings derive from rodent versus human studies. We hope that these additions clarify the conceptual role of these sections within the broader framework of AHN modulation.
- Tables, it is inconsistent to provide a table only for vitamins. Table 2 should be placed earlier in the manuscript if it summarizes all modulators discussed.
Response: We thank the reviewer for this helpful suggestion. The tables have been fully revised to address this concern. The previous vitamin-specific table has been removed, and Table 1 has now been reorganized into a comprehensive summary of all intrinsic and extrinsic factors known to modulate AHN along the dorsoventral axis, ensuring consistency with the structure and focus of the manuscript. Table 2, which relates specifically to the discussion of Alzheimer’s disease, has been retained within the relevant Section 5, as it summarizes only the acquired factors relevant to AD and their relationship to AHN, rather than all modulators covered in the review. Its placement therefore aligns with the flow of the text. We believe these revisions resolve the inconsistencies noted by the reviewer.
- Figure 6, this figure does not address the dorsoventral distinction in the hippocampus and seems unrelated to the main focus.
Response: We thank the reviewer for this observation. The purpose of figure 6 is to provide a concise overview of the specific stages of adult hippocampal neurogenesis (AHN) affected by the various modulatory factors discussed in Section 3. Although it does not show dorsoventral differences, its role is complementary to figures 4 and figure 5, which focus exclusively on regional distinctions along the hippocampal longitudinal axis. Figure 6 instead serves a different function: it illustrates how distinct factors influence different phases of the neurogenic process, helping readers contextualize stage-specific mechanisms summarized throughout the text. Because this information is not redundant with the dorsoventral framework and is directly relevant to the mechanistic discussion in Section 3, we retained the figure but revised its caption and its placement in the manuscript to clarify its purpose and relation to the main narrative. We believe this improves coherence and addresses the reviewer’s concern regarding relevance.
- from reference 34 onward, only author initials are provided
Response: We thank the reviewer for noting this formatting inconsistency. We have corrected the reference list so that all entries follow the required format and include both author surnames and initials. The revised reference section is now consistent throughout.

Reviewer 2 Report
Comments and Suggestions for Authors
The manuscript entitled “Modifying Factors of Adult Hippocampal Neurogenesis: A Dorsoventral Perspective in Health and Disease” presents a very thorough and well written review about the differential effects of several external and internal factors on dorsal and ventral hippocampal neurogenesis.
This is a very well written paper; I enjoy very much reading it. The manuscript is well explained and intelligently organized in sections, headings and subheadings.
I have only some mild concerns that I think are of easy correction.
Concerns:
- The manuscript contains some spelling, formatting, and spacing errors (see page 7). The acronym “ANH” is not explained in the main text, only in the abstract. On line 928, there is an incomplete sentence.
- Please clarify the sentence on lines 140–142: what exactly is not directly modifiable? The circulating hormone levels? The action these hormones exert on the target? Could you reformulate this sentence?
- The title “3.2. Behaviour” does not seem the most appropriate to encompass all the subheadings under it. While “Physical Activity” might be understandable, the subheadings “Sleep,” “Learning,” and “Stress” do not seem well grouped. I suggest “Lifestyle Factors,” but I do not insist on changing it.
- I find the topics “Learning” and “Stress” more ambiguous than they appear. These topics (perhaps more than others) present many confounding factors creating some confusion regarding which parts of the results are causes and which are consequences. Could the authors consider adding a sentence or something indicating that, since both are complex factors, learning or stress per se cannot be solely responsible for the observed responses, as many other factors (some already referenced) strongly influence these outcomes?
- Figure 1 – the representation of the ventral hippocampus in the rat is incorrect. Please review my supplementary PDF file with suggestions and how to correct it.
- Figure 2 – the figure legend does not completely align with the figure; while events during the “Transient increase in CORT levels” are well explained, what happens with MR activation in the case of “Chronic increase in CORT levels” is not explained at all. Could you please clarify this better?
- I do not understand the purpose or necessity of Table 1. Moreover, the table is not referenced in the text. I think Table 1 could be merged with Table 2 to gain significance and relevance. Almost half of the references in Table 1 make little sense. The StatPearls references may be useful for presenting compounds but do not provide information about each one’s action on AHN, which excludes them from Table 1. The references in Table 2 are all relevant and interestingly, addressing the same topic as table 1, the references are different. Either correct and cite Table 1 in the text or merge it with Table 2.
- What do the ** symbols in Table 2 correspond to? They appear in the table but not in any legend or title.
- I also question the relevance of Table 3. What additional information does it contribute to this work? And why is the increased risk only for AD relevant and not for all neuropsychiatric implications?
- Overall, the manuscript is well referenced; the cited references throughout the text support the information presented. However, the references in Table 1 rely too heavily on generic rather than specific references. On the other hand, the reference list is poorly presented, with many references missing volume or page numbers. For example, reference 53: Chaiton JA, W.S., Galea LA, Chronic aromatase inhibition increases ventral hippocampal neurogenesis in middle-aged female mice. Psychoneuroendocrinology, 2019; should be: Chaiton JA, Wong SJ, Galea LA. Chronic aromatase inhibition increases ventral hippocampal neurogenesis in middle-aged female mice. Psychoneuroendocrinology. 2019 Aug;106:111-116.
Reference 197: Martel JL, K.C., Doshi H, et al, Vitamin B1 (Thiamine). StatPearls [Internet], 2024; should be: Martel JL, Doshi H, Sina RE, et al. Vitamin B1 (Thiamine) [Updated 2024 Jan 31]. In: StatPearls [Internet]. Treasure Island (FL): StatPearls Publishing; 2025 Jan-. Available from: https://www.ncbi.nlm.nih.gov/books/NBK482360/.
You need to correct all the references in the References list; a review with questionable references loses credibility.

The manuscript contains some spelling, formatting, and spacing errors
Author Response
Reviewer 2
Comments and Suggestions for Authors
The manuscript entitled “Modifying Factors of Adult Hippocampal Neurogenesis: A Dorsoventral Perspective in Health and Disease” presents a very thorough and well written review about the differential effects of several external and internal factors on dorsal and ventral hippocampal neurogenesis.
This is a very well written paper; I enjoy very much reading it. The manuscript is well explained and intelligently organized in sections, headings and subheadings.
I have only some mild concerns that I think are of easy correction.
Response: We thank the reviewer for the very positive overall assessment of our manuscript and for the constructive suggestions provided. We appreciate the reviewer’s careful reading and helpful comments, which have guided several improvements in clarity and precision. Changes in the revised manuscript are marked as follows: deleted text appears in red with strikethrough, and new or revised text is shown in red.
Concerns:
1. The manuscript contains some spelling, formatting, and spacing errors (see page 7). The acronym “ANH” is not explained in the main text, only in the abstract. On line 928, there is an incomplete sentence.
Response: We thank the reviewer for noting these issues. We have carefully reviewed the manuscript and corrected all spelling, formatting, and spacing errors, including those appearing on page 7. We also now define the acronym AHN (“adult hippocampal neurogenesis”) upon its first use in the main text, not only in the abstract. In addition, the incomplete sentence on line 928 has been corrected. We appreciate the reviewer’s attention to detail.
2. Please clarify the sentence on lines 140–142: what exactly is not directly modifiable? The circulating hormone levels? The action these hormones exert on the target? Could you reformulate this sentence?
Response: We thank the reviewer for pointing out the ambiguity in this sentence. We have reorganized this sentence to clarify that circulating hormone levels are not directly regulated by the hippocampal neurogenic area, whereas local sensitivity to hormonal signals (e.g., receptor density, downstream signaling pathways) can be modulated by physiological and experiential factors. This revised wording removes the ambiguity and improves interpretability.
3. The title “3.2. Behaviour” does not seem the most appropriate to encompass all the subheadings under it. While “Physical Activity” might be understandable, the subheadings “Sleep,” “Learning,” and “Stress” do not seem well grouped. I suggest “Lifestyle Factors,” but I do not insist on changing it.
Response: We thank the reviewer for this helpful suggestion. We agree that the previous title “Behavior” did not optimally encompass the range of topics included in the subsection. We have therefore revised the heading to “Lifestyle and Environmental Factors” to better reflect the content of the subheadings (“Physical Activity,” “Sleep,” “Learning,” and “Stress”). We appreciate the reviewer’s thoughtful recommendation.
4. I find the topics “Learning” and “Stress” more ambiguous than they appear. These topics (perhaps more than others) present many confounding factors creating some confusion regarding which parts of the results are causes and which are consequences. Could the authors consider adding a sentence or something indicating that, since both are complex factors, learning or stress per se cannot be solely responsible for the observed responses, as many other factors (some already referenced) strongly influence these outcomes?
Response: We thank the reviewer for this insightful comment. We agree that both learning and stress represent complex, multifaceted states that engage numerous physiological and behavioral processes, making it difficult to attribute neurogenic effects to these factors in isolation. In the revised manuscript, we have added clarifying statements in the corresponding subsections noting that learning- and stress-related changes in AHN likely reflect the combined influence of multiple interacting mechanisms (e.g., reward signaling, emotional arousal, glucocorticoid dynamics, environmental context), rather than the isolated effect of “learning” or “stress” alone. We believe these additions improve clarity and appropriately qualify the interpretation of these findings.
5. Figure 1 – the representation of the ventral hippocampus in the rat is incorrect. Please review my supplementary PDF file with suggestions and how to correct it.
Response: We thank the reviewer for this important observation. The representation of the ventral hippocampus in the rat brain in the previous version was indeed oversimplified and did not accurately reflect its anatomical position. Following the reviewer’s helpful supplementary suggestions, we have replaced the illustration in figure 1 with a corrected and more anatomically accurate depiction. In addition, we have updated the hippocampal representation in figure 5 to ensure consistency across the manuscript. We appreciate the reviewer’s guidance in improving the accuracy of these figures.
6. Figure 2 – the figure legend does not completely align with the figure; while events during the “Transient increase in CORT levels” are well explained, what happens with MR activation in the case of “Chronic increase in CORT levels” is not explained at all. Could you please clarify this better?
Response: We thank the reviewer for this helpful observation. We agree that the original legend did not adequately explain receptor dynamics during chronic CORT elevations. In the revised version, we have clarified that under sustained high CORT levels, MRs become saturated early, and the predominant signaling shifts to GR overactivation, which drives the suppressive effects on AHN. The legend has been updated to explicitly describe this transition from MR engagement (during transient elevations) to GR-predominant signaling (during chronic elevations). We believe this clarification now aligns the legend more accurately with the events illustrated in the figure.
7. I do not understand the purpose or necessity of Table 1. Moreover, the table is not referenced in the text. I think Table 1 could be merged with Table 2 to gain significance and relevance. Almost half of the references in Table 1 make little sense. The StatPearls references may be useful for presenting compounds but do not provide information about each one’s action on AHN, which excludes them from Table 1. The references in Table 2 are all relevant and interestingly, addressing the same topic as table 1, the references are different. Either correct and cite Table 1 in the text or merge it with Table 2.
Response: We thank the reviewer for this thoughtful and detailed feedback regarding Table 1. In revising the manuscript, we recognized that the previous version of Table 1 did not clearly serve the central purpose of the review and could be misleading in its scope and referencing. We therefore replaced Table 1 entirely with a new table summarizing all factors that have been studied with respect to dorsoventral differences in AHN, which aligns more directly with the main theme of the manuscript. The revised Table 1 now includes only modulators for which dorsoventral data exist, uses a consistent referencing structure, and is fully integrated into the discussion. We have also ensured that the table is explicitly cited in the main text. We believe that this redesign addresses the reviewer’s concerns and substantially improves the clarity and relevance of the table.
8. What do the ** symbols in Table 2 correspond to? They appear in the table but not in any legend or title.
Response: We thank the reviewer for noticing this oversight. The “**” symbols in Table 2 were not explained in the previous version. In the revised manuscript, we have added a clear explanation in the table legend specifying the meaning of each symbol. We appreciate the reviewer bringing this to our attention.
9. I also question the relevance of Table 3. What additional information does it contribute to this work? And why is the increased risk only for AD relevant and not for all neuropsychiatric implications?
Response: We thank the reviewer for this helpful comment. In the revised manuscript, the former Table 3 has been renumbered as Table 2 following the removal of the previous Table 1. We agree that the earlier version of this table did not clearly convey its relevance to the dorsoventral framework of the review. To improve clarity, we have revised Table 2 so that the “Effect on AHN” column now specifies whether each factor predominantly affects dorsal or ventral neurogenesis, thereby aligning the table more directly with the central theme of dorsoventral specialization. We have also clarified in the main text (Section 5.3, 2nd paragraph) in the revised manuscript) why this table focuses on Alzheimer’s disease (AD): several modulators that exhibit region-specific effects on AHN have also been implicated in AD risk or resilience, suggesting a potential mechanistic link between dorsoventral neurogenesis and AD-related vulnerability. Together, these revisions strengthen the relevance and interpretability of Table 2 within the overall structure of the manuscript.
10. Overall, the manuscript is well referenced; the cited references throughout the text support the information presented. However, the references in Table 1 rely too heavily on generic rather than specific references. On the other hand, the reference list is poorly presented, with many references missing volume or page numbers. For example, reference 53: Chaiton JA, W.S., Galea LA, Chronic aromatase inhibition increases ventral hippocampal neurogenesis in middle-aged female mice. Psychoneuroendocrinology, 2019; should be: Chaiton JA, Wong SJ, Galea LA. Chronic aromatase inhibition increases ventral hippocampal neurogenesis in middle-aged female mice. Psychoneuroendocrinology. 2019 Aug;106:111-116.
Reference 197: Martel JL, K.C., Doshi H, et al, Vitamin B1 (Thiamine). StatPearls [Internet], 2024; should be: Martel JL, Doshi H, Sina RE, et al. Vitamin B1 (Thiamine) [Updated 2024 Jan 31]. In: StatPearls [Internet]. Treasure Island (FL): StatPearls Publishing; 2025 Jan-. Available from: https://www.ncbi.nlm.nih.gov/books/NBK482360/.
You need to correct all the references in the References list; a review with questionable references loses credibility.
Response: We thank the reviewer for this important observation. We have thoroughly revised the entire reference list to ensure accuracy, completeness, and consistency with journal guidelines. The specific examples noted by the reviewer have been corrected, and we extended this correction process to all references in the manuscript. In addition, because the earlier version of Table 1 (now removed) relied partly on generic references, the revised Table 1 now includes only specific, peer-reviewed primary sources directly relevant to dorsoventral AHN. Accordingly, references that were linked solely to the removed table, such as Martel et al. (StatPearls), no longer appear in the revised manuscript. We thank the reviewer for emphasizing the importance of reference quality, which has helped us substantially improve the rigor and consistency of the manuscript.
Also, we have carefully reviewed the entire manuscript and corrected all spelling, formatting, and spacing errors throughout the text.

Reviewer 3 Report
Comments and Suggestions for Authors
I appreciate the opportunity to review this very interesting article. It was expertly written and the figures and tables were well developed and helped summarize and convey the information that was provided. This was so well written, I would recommend using it for an advanced neuroscience class. Although it was very long, it was very thorough and it was a page turner. It was a joy to read. I do a have a few comments and suggestions, but overall, I don’t have many criticisms.
- Lines 120-123 – “new neurons in this phase have a lower threshold for long term potentiation….” I get this but can the authors explain why more precisely… Also, do new neurons also exhibit “higher synaptic plasticity”? Don’t they need to have a higher threshold for long term potentiation to have higher synaptic plasticity?
- There are many gems in this article that I was glad to encounter as I just don’t have the time to do the thorough literature search such as this (lines 153-154) – “MR and GR and abundantly expressed in hippocampal regions, with MRs exhibiting ~10-fold higher affinity for CORT than GRs.”
- Line 165 – The role of “sexual experience” on elevated neurogenic rates and higher circulating CORT levels really opens up a new way of thinking of about sex as being healthy for brain health.
- Lines 168-170 – I agree with the other authors that there is probably more a U-shaped curvilinear relationship with the level of CORT.
- Line 241 – I think the authors mean “long-term” and not “long-derm”.
- Period is missing in Line 254.
- Lines 329 – Not sure what “manganese toxicity” (I did look it up). It might be helpful to put in parentheses what this is exactly.
- The whole theme of the article is summarized in lines 510-513. I am glad this is repeated elsewhere to reinforce.
- Loved Figure 5.
- Remove “by” at the end of the sentence – LINE 928
- Loved Figure 6.
Author Response
Reviewer 3
Comments and Suggestions for Authors
I appreciate the opportunity to review this very interesting article. It was expertly written and the figures and tables were well developed and helped summarize and convey the information that was provided. This was so well written, I would recommend using it for an advanced neuroscience class. Although it was very long, it was very thorough and it was a page turner. It was a joy to read. I do a have a few comments and suggestions, but overall, I don’t have many criticisms.
Response: We thank the reviewer for their very positive assessment of our manuscript and for the encouraging comments. We are pleased that the reviewer found the article clear, thorough, and useful. We have addressed the specific suggestions below and revised the manuscript accordingly. Changes in the revised manuscript are marked as follows: deleted text appears in red with strikethrough, and new or revised text is shown in red.
- Lines 120-123 – “new neurons in this phase have a lower threshold for long term potentiation….” I get this but can the authors explain why more precisely… Also, do new neurons also exhibit “higher synaptic plasticity”? Don’t they need to have a higher threshold for long term potentiation to have higher synaptic plasticity?
Response: We thank the reviewer for raising this important point. We have clarified the explanation in the manuscript. Immature granule neurons exhibit increased intrinsic excitability and reduced inhibitory tone, which results in a lower threshold for inducing LTP. A lower threshold means that LTP can be triggered by weaker or more physiologically relevant inputs, which is precisely what underlies their higher synaptic plasticity. These properties allow them to integrate into existing circuits with greater flexibility than mature neurons. Thus, there is no contradiction between “lower threshold for LTP” and “higher synaptic plasticity”; the former is one of the mechanisms enabling the latter. We have expanded the relevant sentence in Section 2 to reflect this mechanistic basis.
- There are many gems in this article that I was glad to encounter as I just don’t have the time to do the thorough literature search such as this (lines 153-154) – “MR and GR and abundantly expressed in hippocampal regions, with MRs exhibiting ~10-fold higher affinity for CORT than GRs.”
Response: We sincerely thank the reviewer for this very encouraging comment. We are grateful that the reviewer found the manuscript informative and that specific details, such as the difference in MR and GR affinity for corticosterone, were useful. We appreciate the recognition of the effort involved in synthesizing this broad literature.
- Line 165 – The role of “sexual experience” on elevated neurogenic rates and higher circulating CORT levels really opens up a new way of thinking of about sex as being healthy for brain health.
Response: We thank the reviewer for this thoughtful and encouraging remark. We agree that the findings regarding sexual experience, transient CORT elevations, and enhanced neurogenesis offer an intriguing perspective on how natural rewarding behaviors can promote hippocampal plasticity. We are pleased that this aspect of the literature resonated with the reviewer.
- Lines 168-170 – I agree with the other authors that there is probably more a U-shaped curvilinear relationship with the level of CORT.
Response: We thank the reviewer for highlighting this point. We agree that the available evidence supports a U-shaped relationship between circulating CORT levels and AHN, with both insufficient and excessive glucocorticoid signaling exerting suppressive effects. This interpretation is consistent with the mechanistic model presented in our manuscript. We have slightly refined the wording in this section to emphasize the curvilinear nature of the relationship.
- Line 241 – I think the authors mean “long-term” and not “long-derm”.
Response: We thank the reviewer for noticing this typographical error. We have corrected “long-derm” to “long-term” in the revised manuscript.
- Period is missing in Line 254.
Response: We thank the reviewer for noting this formatting oversight. We have added the missing period at Line 254 in the revised manuscript.
- Lines 329 – Not sure what “manganese toxicity” (I did look it up). It might be helpful to put in parentheses what this is exactly.
Response: We thank the reviewer for this helpful suggestion. We have clarified the term “manganese toxicity” in the revised manuscript by briefly indicating that it refers to excessive manganese exposure leading to oxidative stress and neurotoxic effects. This should assist readers who are less familiar with this literature.
- The whole theme of the article is summarized in lines 510-513. I am glad this is repeated elsewhere to reinforce.
Response: We thank the reviewer for this encouraging comment. We are pleased that the thematic summary in lines 510–513 clearly conveyed the central message of the manuscript and that its reinforcement elsewhere in the text was effective.
- Loved Figure 5.
Response: We sincerely thank the reviewer for this very kind comment. We are delighted that Figure 5 was appreciated and that it effectively conveyed the concepts we intended to highlight.
- Remove “by” at the end of the sentence – LINE 928
Response: We thank the reviewer for noting this wording issue. We have removed the extraneous “by” at Line 928 in the revised manuscript.
- Loved Figure 6.
Response: We sincerely thank the reviewer for this generous comment. We are delighted that Figure 6 was well received and that it effectively captured the concepts we aimed to illustrate. We greatly appreciate the reviewer’s encouraging feedback.

Round 2
Reviewer 1 Report
Comments and Suggestions for Authors
The manuscript by Ioannis Erginousakis and Costas Papatheodoropoulos has much and significantly improved.
I have a minor comment, please check for 'neurogenic process' and write 'neurogenesis' instead (such as lines 780, 1163, 1193 etc.); for 'neurogenic pool', i.e., line 560, do authors mean 'proliferation' or 'progenitor pool', cells which to a high degree follow a neuronal fate.
Author Response
Reviewer 1
I have a minor comment, please check for 'neurogenic process' and write 'neurogenesis' instead (such as lines 780, 1163, 1193 etc.); for 'neurogenic pool', i.e., line 560, do authors mean 'proliferation' or 'progenitor pool', cells which to a high degree follow a neuronal fate.
We thank the reviewer for this careful observation. We have systematically replaced the term “neurogenic process” with “neurogenesis” throughout the manuscript to ensure terminological consistency and precision. With regard to the term “neurogenic pool”, we agree that this wording could be ambiguous. We replaced the term “neurogenic progenitor pool” with “proliferating progenitor pool” to avoid ambiguity and to reflect more accurately the interpretation of Klempin et al. (J Neurosci, 2013), which demonstrates that exercise primarily modulates progenitor proliferation.